# *Legionella* effector *Lp*PIP recruits protein phosphatase 1 to the mitochondria to induce dephosphorylation of outer membrane proteins

Kai-Qi Yek[1], Evie R. Hodgson[1], Ching-Seng Ang[2], Catherine S. Palmer[1], Ann E. Frazier[3,4], Hayley J. Newton[5]*, Diana Stojanovski[1]*

1 Department of Biochemistry and Pharmacology, The University of Melbourne, Parkville, Victoria, Australia, 2 The Bio21 Molecular Science and Biotechnology Institute, The University of Melbourne, Parkville, Victoria, Australia, 3 Murdoch Children's Research Institute, Royal Children's Hospital, Parkville, Victoria, Australia, 4 Department of Paediatrics, The University of Melbourne, Parkville, Victoria, Australia, 5 Infection Program, Monash Biomedicine Discovery Institute and Department of Microbiology, Monash University, Clayton, Victoria, Australia

* hayley.newton@monash.edu (HJN); d.stojanovski@unimelb.edu.au (DS)

## Abstract

*Legionella pneumophila* utilizes a type IVB secretion system (T4SS) to translocate over 300 effector proteins into host cells, hijacking cellular processes, including those within the mitochondrion. Currently, no *Legionella* effectors have been identified at the mitochondrial outer membrane, a critical interface between the organelle and the rest of the cell. We screened the *Legionella* effector repertoire for features of mitochondrial tail-anchored (TA) proteins and identified four putative TA effectors. Among them, *Lp*PIP (Lpg1625) localizes to the mitochondrial outer membrane and interacts with all three isoforms of protein phosphatase 1 (PP1) via an RVxF motif, functioning as a PP1-interacting protein (PIP). Importantly, PP1 remains catalytically active upon interaction with *Lp*PIP to dephosphorylate mitochondrial outer membrane proteins. Altering the TA signature to direct *Lp*PIP to the ER induces ER-recruitment of PP1 and dephosphorylation of ER-resident proteins, indicating that *Lp*PIP controls PP1 localization and not substrate specificity. This study uncovers a novel pathogen-mediated strategy to modulate PP1 and manipulate the host cell phosphoproteome.

## Introduction

*Legionella pneumophila* is a gram-negative bacterium found ubiquitously in aquatic environments, with a wide variety of free-living eukaryotic phagotrophs, mainly amebae, as its natural reservoir [1]. Through co-evolution with its eukaryotic hosts, *L. pneumophila* has evolved the capacity to "accidentally" infect human alveolar macrophages when inhaled, resulting in the severe pneumonia known as Legionnaires' disease [2,3]. Throughout its intracellular lifecycle, a highly sophisticated Dot/Icm type IVB secretion system (T4SS) is required to translocate over 300 effector proteins into

**Data availability statement:** The mass spectrometry proteomics data have been deposited to the ProteomeXchange Consortium via the PRIDE partner repository with the dataset identifier PXD057005. All other data is presented within the figures and supp figures and files of this manuscript.

**Funding:** This work was funded by the Australian Research Council. Grant DP240101332 awarded to both DS and HN. E.H is funded through DP240101332. K.Q.Y. is supported by Melbourne International Fee Remission Scholarship (MIFRS) and Melbourne International Research Scholarship (MIRS). The funders did not play any role in the study design, data collection and analysis, decision to publish, or preparation of the manuscript.

**Competing interests:** The authors have declared that no competing interests exist.

**Abbreviations:** ABC, ammonium bicarbonate; BCYE, Buffered-Charcoal Yeast Extract; BN-PAGE, Blue native polyacrylamide gel electrophoresis; ER, endoplasmic reticulum; FUNDC2, FUN14 domain containing 2; GO, gene ontology; *Lp*PIP, *Legionella pneumophila* PP1-Interacting Protein; MFF, mitochondrial fission factor; MOI, multiplicity of infection; PIP, PP1-interacting protein; PK, Proteinase K; PP1, protein phosphatase 1; pSer/pThr, phosphoserine/phosphothreonine; ROI, region of interest; RPMI, Roswell Park Memorial Institute; SDS, sodium dodecyl sulfate; TA, tail-anchored; TMD, transmembrane domain; T4SS, type IVB secretion system.

host cells [4–6]. Effector proteins manipulate a wide array of host cellular processes, facilitating the pathogen's intracellular survival and replication.

Among the targets of *L. pneumophila* effector proteins is the mitochondrion, an organelle central to energy production, apoptosis, immune signaling, calcium homeostasis, and lipid metabolism [7–10]. Given the integral role of mitochondria in cellular homeostasis, it is a common target for manipulation by bacterial pathogens, including *L. pneumophila, Mycobacterium tuberculosis, Listeria monocytogenes, Shigella flexneri*, and *Coxiella burnetii* [11–15].

To date, numerous *L. pneumophila* effector proteins have been shown to target the mitochondrion and modulate its activity. These include: (i) MitF (Lpg1976), which promotes mitochondrial fragmentation by recruiting the host fission protein Drp1 to the mitochondrial surface, resulting in a cellular metabolic shift to glycolysis [10,16]; (ii) Lpg2444, which has been implicated in preventing mitochondrial fragmentation [17]; (iii) LncP (Lpg2905), an inner membrane-localized ATP transporter [10,18]; (iv) LegS2/*Lp*Spl (Lpg2176), which maintains mitochondrial membrane potential by modulating the activity of the mitochondrial $F_oF_1$-ATPase [19–22]; (v) Ceg3 (Lpg0080) and Lpg0081, which add or remove ADP-ribose on ADP/ATP translocases, respectively, to modulate ATP/ADP transport across the mitochondrial inner membrane [23,24]; and (vi) the serine protease Lpg1137, which localizes to the mitochondrial-associated membrane and degrades syntaxin-17, blocking autophagy and apoptosis [25]. These findings showcase the sophisticated strategies *L. pneumophila* utilizes to manipulate mitochondrial function in the host cell.

Mitochondrial biology is intricately linked to the organelle's double membrane structure, which partitions it into four sub-compartments: the outer membrane, inter-membrane space, inner membrane, and matrix. This compartmentalization supports the distribution of macromolecules and specializes activities within the organelle [26]. The mitochondrial outer membrane upholds organelle integrity and regulates key processes, including apoptosis, mitophagy, organelle dynamics, protein biogenesis, metabolites transport, and innate immune signaling [27–30]. Despite these critical functions, there are currently no described *L. pneumophila* effector proteins that localize to the mitochondrial outer membrane.

Integral proteins of the mitochondrial outer membrane include β-barrel proteins, signal-anchored proteins, C-tail anchored (TA) proteins, and polytopic membrane proteins [31]. TA proteins are found in mitochondria as well as in organelles of the endomembrane system and peroxisomes [32]. They possess an N-terminal cytosolic domain, a C-terminal transmembrane domain (TMD) that anchors the protein to a specific membrane, and a short luminal C-terminal tail [32]. TA proteins targeted to mitochondria often exhibit a net positive charge within their C-terminal tails [33]. Mitochondrial TA proteins are involved in diverse functions, including apoptosis, mitochondrial dynamics, and mitochondrial protein biogenesis [33]. Given this, we queried whether *L. pneumophila* encodes effectors that target the mitochondrial outer membrane via TA mechanisms.

Using a bioinformatics approach, we identified four *L. pneumophila* T4SS effectors, Lpg1625, Lpg1803, Lpg2344, and Lpg2444, with characteristics of mitochondrial

TA proteins. We show that Lpg1625 is a true TA protein that anchors to the mitochondrial outer membrane via a C-terminal TMD and positively charged residues surrounding the TMD. We show that Lpg1625 interacts with human protein phosphatase 1 (PP1), a highly conserved serine/threonine phosphatase involved in regulating numerous cellular processes [34]. The recruitment of PP1 to mitochondria relies on a specific RVxF motif in Lpg1625 and modulates the host phosphoproteome through dephosphorylation of mitochondrial outer membrane proteins. Lpg1625 behaves as a PP1-interacting protein (PIP), and we therefore accordingly assign the name *Legionella pneumophila* PP1-Interacting Protein (*Lp*PIP) to this effector. This study uncovers the ability of *L. pneumophila* to modulate the host mitochondrial phosphoproteome by hijacking a crucial cellular phosphatase, PP1.

## Results

### *L. pneumophila* targets mitochondria using T4SS effectors with tail anchors

To identify *L. pneumophila* T4SS effectors with characteristics of mitochondrial TA proteins, we conducted a bioinformatic screen of 302 experimentally confirmed or putative *L. pneumophila* T4SS effector proteins (Fig 1A and S1 Text) [4,35,36]. Using TMHMM and TOPCONS, 22 effectors were predicted to contain a single TMD [37,38]. Of these, five had a TMD within 30 residues of the stop codon, and four (Lpg1625, Lpg1803, Lpg2344, and Lpg2444) exhibited a positively charged C-terminal tail, a hallmark of mitochondrial TA proteins (Fig 1B and S1 Table) [33]. To validate the mitochondrial localization of the four predicted TA proteins, N-terminally FLAG-tagged effectors were transiently expressed in HeLa cells and analyzed by immunofluorescence microscopy. All four proteins showed colocalization with the mitochondrial marker NDU-FAF2, confirming mitochondrial localization (Fig 1C). Lpg1625 and Lpg2444 have been shown to localize to mitochondria previously, with Lpg1625 suggested to induce caspase 3 activation [39], and Lpg2444 suggested to prevent mitochondrial fragmentation [17]. However, detailed functional characterization of these effectors has not been undertaken. Lpg2344 was reported to have a vesicular localization [40], and Lpg1803 has not been functionally characterized. In this study, we selected Lpg1625 as the first candidate for further functional analysis.

### Lpg1625 is a mitochondrial tail-anchored (TA) effector

Deletion of *lpg1625* did not influence the ability of *L. pneumophila* to replicate in THP-1 macrophage-like cells (S1 Fig and S2 Table). However, it may contribute to the intracellular success of *L. pneumophila* under specific circumstances and/or in a manner that is functionally redundant with other T4SS effectors [6]. We ectopically expressed FLAGLpg1625 in HeLa or HEK cells to facilitate downstream biochemical characterization. First, we confirmed that FLAGLpg1625 localizes exclusively to mitochondria and not to other organelles typically targeted by TA proteins (S2A–S2D Fig) [32]. We then biochemically validated its localization to the mitochondrial outer membrane using sub-fractionation analysis and carbonate extraction (Fig 1D). Mitochondria isolated from HeLa cells transfected with FLAGLpg1625 were either left untreated, subjected to hypo-osmotic swelling, or solubilized in Triton X-100, followed by treatment with Proteinase K (PK) (Fig 1D). FLAGLpg1625 was accessible to external protease in intact mitochondria, mirroring the outer membrane marker Mfn2, consistent with an outer membrane localization (Fig 1D, lanes 1–2). Sodium carbonate extraction recovered FLAGLpg1625 in the pellet fraction, suggesting the protein is membrane integrated (Fig 1D, lanes 7–8). Based on this profile, we concluded Lpg1625 is integrated into the mitochondrial outer membrane via a C-terminal TMD.

To confirm the TA characteristics of Lpg1625, we generated truncation and charge mutants and examined their subcellular localization by immunofluorescence analysis (Fig 1E). Constructs included: FLAGLpg1625$^{1-108}$ (cytosolic domain), FLAGLpg1625$^{85-130}$ (TMD and C-tail), FLAGLpg1625$^{K128A}$ (single charge mutant), and FLAGLpg1625$^{AAA}$ (K97A/R100A/K128A triple charge mutant) (Fig 1E). These experiments demonstrated that the TMD and C-tail of Lpg1625 are sufficient for mitochondrial targeting of Lpg1625 (Fig 1E). The K128A and K97A/R100A/K128A mutants, designed to remove positive charge flanking the TMD, showed mislocalization to the endoplasmic reticulum (ER) (Figs 1E and S2E), consistent with previous

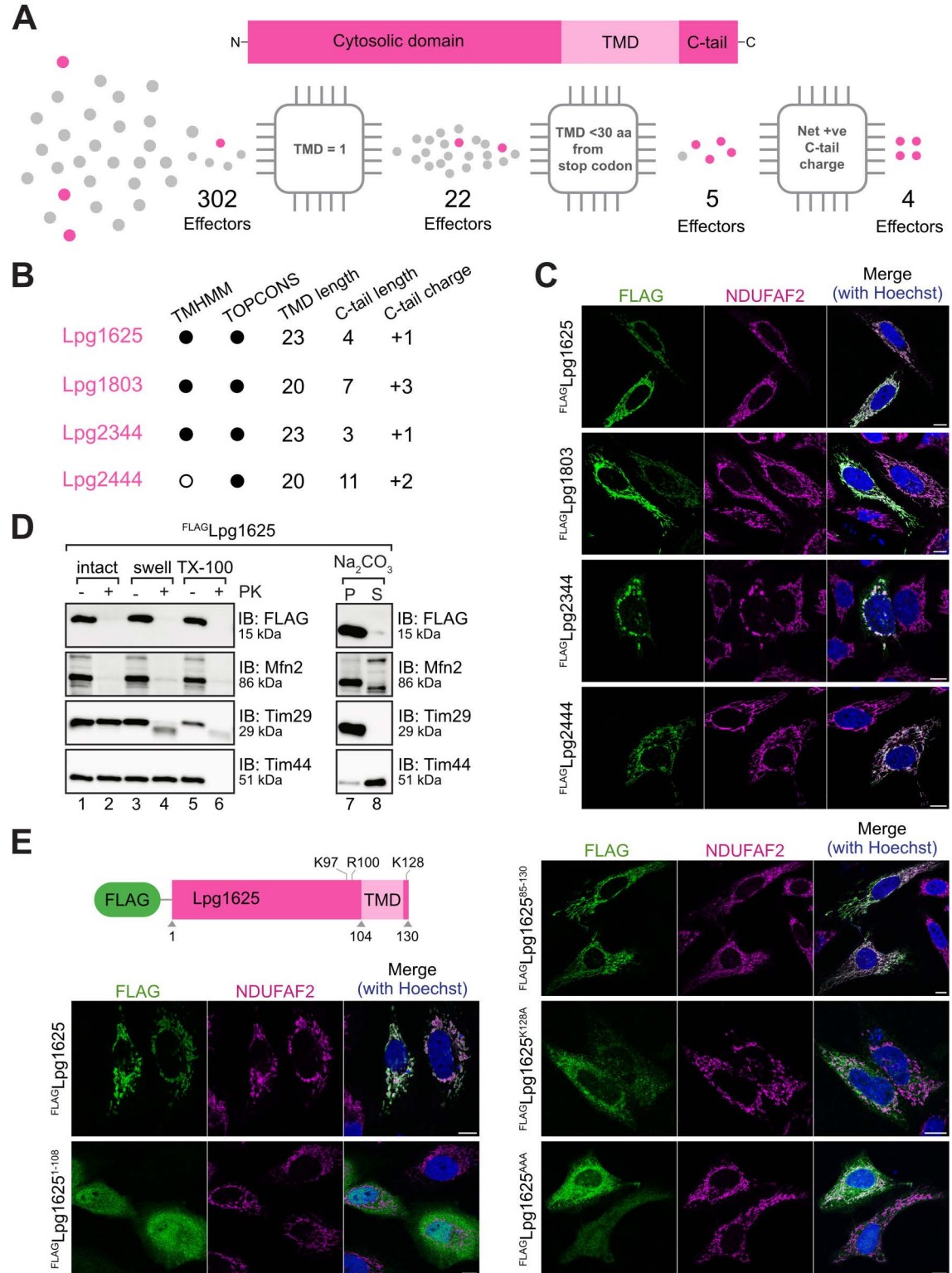

**Fig 1. *L. pneumophila* targets mitochondria using T4SS effectors with tail anchors (TA). (A)** Schematic representation of the bioinformatics work-flow used to screen for *L. pneumophila* T4SS effectors (see S1 Text) that contain a TA with net positive tail charges (see S1 Table). **(B)** Features of the four effectors predicted to contain a TA with a net positive charge in their C-tail. TMHMM and TOPCONS were used to predict the number and location of

transmembrane domains (TMDs) present within the 302 effectors screened. A black dot indicates the prediction of one TMD, while a white dot indicates no predicted TMD. The length of the predicted TMD and C-tail, along with the net C-tail charge, is indicated. **(C)** Representative images of HeLa cells transiently transfected with FLAGLpg1625, FLAGLpg1803, FLAGLpg2344, or FLAGLpg2444. Transfected cells were fixed and processed for immunofluorescence with antibodies against FLAG (green) and NDUFAF2 (mitochondria; magenta). Nuclei were stained with Hoechst 33258 (blue). Scale bar represents 10 µm. **(D)** Mitochondrial sub-fractionation and carbonate extraction using mitochondria isolated from HeLa cells transiently transfected with FLAGLpg1625. Mitochondria were either left intact (lanes 1 and 2), subjected to hypotonic swelling (lanes 3 and 4), or solubilized with 0.5% Triton X-100 (lanes 5 and 6). Samples treated with Proteinase K (PK) are indicated (+). Mitochondria were also subjected to sodium carbonate extraction, with pellet (P) and supernatant (S) fractions (lanes 7 and 8) separated via ultracentrifugation. Samples were analyzed using SDS-PAGE and immunoblotting using the indicated antibodies. Corresponding raw images are available in the Supporting information (S1 Raw Images). **(E)** HeLa cells transiently transfected with FLAGLpg1625, FLAGLpg1625[1-108], FLAGLpg1625[85-130], FLAGLpg1625[K128A], and FLAGLpg1625[AAA] (K97A/R100A/K128A) were processed for immunofluorescence with antibodies against FLAG (green) and NDUFAF2 (mitochondria; magenta). Nuclei were stained with Hoechst 33258 (blue). Scale bar represents 10 µm.

findings [41]. We confirmed FLAGLpg1625[85-130] (TMD and C-tail) was also sufficient for membrane integration using carbonate extraction (S2F Fig). Taken together, this suggests that the *L. pneumophila* effector repertoire contains TA proteins, and we have biochemically confirmed that Lpg1625 is a *bona fide* mitochondrial outer membrane TA effector protein.

## Lpg1625 interacts with protein phosphatase 1 (PP1)

We sought to elucidate the functional role of Lpg1625 by identifying its interacting partners using co-immunoprecipitation and mass spectrometry. This revealed the three isoforms of PP1 (PPP1CA, PPP1CB, and PPP1CC), as the top interactors of Lpg1625, detected with 4, 7, and 5 unique peptides, respectively (Fig 2A and S3 Table). This suggests a novel and functional role for all three PP1 isoforms in conjunction with Lpg1625 at the mitochondrial surface. Additional co-immunoprecipitated proteins included MTCH1 and MTCH2 which may facilitate membrane integration of Lpg1625 [31,42]. The VDAC proteins were also co-immunoprecipitated, likely due to their abundance and physical proximity to Lpg1625 in the mitochondrial outer membrane (Fig 2A). To rule out potential artifacts, we generated a stable tetracycline-inducible Flp-In T-REx 293 cell line expressing FLAGLpg1625, allowing regulation of expression to low levels (S3A Fig). Sub-fractionation analysis of mitochondria isolated from these cells confirmed the mitochondrial outer membrane profile of FLAGLpg1625 (S3B Fig). Co-immunoprecipitation using this stable cell line confirmed PP1 as the primary interactor of Lpg1625, even at low expression levels (S3C Fig and S4 Table).

PP1 is a ubiquitously expressed serine/threonine (Ser/Thr) phosphatase, which has a broad substrate specificity and is responsible for a significant proportion of dephosphorylation events in eukaryotic cells [45]. It contains three independent catalytic subunits encoded by three different genes, namely, *PPP1CA*, *PPP1CB*, and *PPP1CC* [46], with 85% sequence identity among them. These proteins contain highly conserved catalytic domains but diverge at the N- and C-termini [46,47]. PP1 functions as a holoenzyme through interaction with PIPs, a diverse group of regulatory proteins that modulate PP1 activity, substrate specificity, and subcellular localization [45,48]. We hypothesized that Lpg1625 acts as a PIP by interacting with all three catalytic subunits of PP1 and recruiting them to the mitochondrial outer membrane, thereby modulating the host mitochondrial phosphoproteome.

## Lpg1625 recruits PP1 to mitochondria via its RVxF motif

PIPs bind PP1 typically through short linear motifs such as RVxF, which is found in approximately 70% of known PIPs [45,49,50]. Examining the Lpg1625 protein sequence revealed the presence of a RVxF motif within its N-terminal cytosolic domain (amino acids 17–21), with two lysine residues upstream of the motif (at position 15 and 16), which can increase affinity for PP1 (Fig 2B) [51]. To investigate the functional importance of this motif in the interaction between Lpg1625 and PP1, we introduced alanine substitutions at two key residues of the RVxF motif of Lpg1625 (K17A and V19A) to create the mutant construct FLAGRVxF_dead [51]. While FLAGRVxF_dead localized correctly to the mitochondrial outer membrane in both transient (Fig 2C) and stable expression systems (S3D Fig), co-immunoprecipitation experiments demonstrated a complete

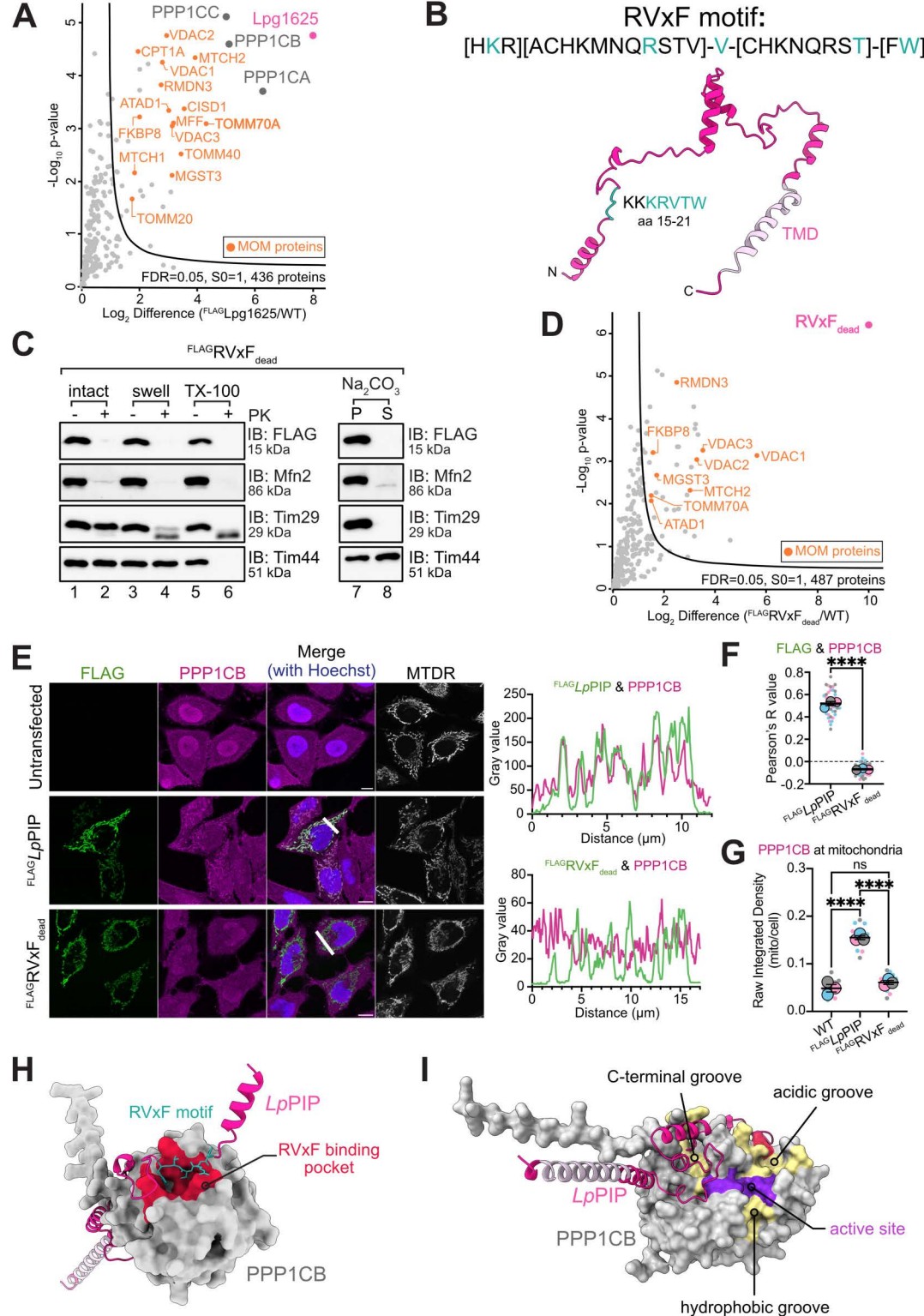

**Fig 2. Lpg1625 recruits PP1 to mitochondria via an RVxF motif. (A)** Mitochondria were isolated from wild-type HeLa cells or cells transiently transfected with FLAGLpg1625 and solubilized in 1% digitonin prior to immunoprecipitation with anti-FLAG conjugated agarose beads. Eluates were analyzed by mass spectrometry. The $\log_2$ fold change of the mean LFQ intensity is plotted against $-\log_{10}$ p-value ($n = 3$ technical replicates). The curve indicates

significantly enriched proteins (FDR = 0.05, s0 = 1). Mitochondrial outer membrane proteins are labeled in orange with the annotation from MitoCarta 3.0. Corresponding data are available in the Supporting information (S3 Table). **(B)** AlphaFold [43] was used to predict the structure of Lpg1625, which was labeled with the RVxF motif and transmembrane domain (TMD) using ChimeraX [44]. The model has an average per-residue model confidence score (pLDDT) of 65.94, indicating low model confidence. **(C)** Mitochondria sub-fractionation and carbonate extraction were performed using mitochondria isolated from HeLa cells transiently transfected with $^{FLAG}$RVxF$_{dead}$. Mitochondria were either left intact (lanes 1 and 2), subjected to hypotonic swelling (lanes 3 and 4), or solubilized with 0.5% Triton X-100 (lanes 5 and 6). Samples treated with Proteinase K (PK) are indicated (+). For carbonate extraction, mitochondria were washed in a fresh solution of 0.1 M sodium carbonate and integral (pellet, P) and soluble (supernatant, S) proteins were separated using ultracentrifugation. Samples were analyzed using SDS-PAGE and immunoblotting using the indicated antibodies. Corresponding raw images are available in the Supporting information (S1 Raw Images). **(D)** Mitochondria isolated from wild-type HeLa cells or cells transiently transfected with $^{FLAG}$RVxF$_{dead}$ were solubilized in 1% digitonin buffer prior to immunoprecipitation with anti-FLAG conjugated agarose beads. Eluates were processed for mass spectrometry. The Log$_2$ fold change of mean LFQ intensity is plotted against −Log$_{10}$ p-value (n = 3 technical replicates). The curve indicates significantly enriched proteins (FDR = 0.05, s0 = 1). Mitochondrial outer membrane proteins are labeled in orange with the annotation from MitoCarta3.0. Corresponding data are available in the Supporting Information (S3 Table). **(E)** HeLa cells expressing $^{FLAG}$LpPIP, or $^{FLAG}$RVxF$_{dead}$ were stained with 100 nM MitoTracker Deep Red FM (mitochondria; MTDR) and, following fixation, processed for immunofluorescence using antibodies against FLAG (green) and PPP1CB (magenta). Nuclei were stained with Hoechst 33258 (blue). Scale bar represents 10 μm. Fluorescence signal profiles of FLAG and PPP1CB were measured across the indicated line of interest. Corresponding raw data are available in the Supporting information (S1 Data). **(F)** The correlation between PPP1CB and $^{FLAG}$LpPIP was assessed using the Pearson correlation coefficient (r) in HeLa cells expressing $^{FLAG}$LpPIP or $^{FLAG}$RVxF$_{dead}$ (n = 3 independent experiments, 45 cells in total). Data represent mean ± SEM of the three experiments represented in three different colors. Larger circles represent the mean values from each experiment, while smaller circles are values from individual cells. An unpaired t test on the means revealed a significant difference of 0.59 ± 0.02 between groups (p-value < 0.0001). Corresponding raw data are available in the Supporting information (S1 Data). **(G)** Proportion of PPP1CB localized to mitochondria in untransfected and HeLa cells transiently transfected with $^{FLAG}$LpPIP or $^{FLAG}$RVxF$_{dead}$, was calculated using raw integrated density in Fiji. Representative images are shown in Fig 2E. Data are presented as mean ± SEM (n = 3 independent experiments, 5 cells each). Ordinary one-way ANOVA showed no significant difference between the groups. Corresponding raw data are available in the Supporting information (S1 Data). **(H)** AlphaFold3 predicted structure of LpPIP-PPP1CB complex (ipTM = 0.86; pTM = 0.85) labeled with the RVxF motif on LpPIP (green) and known RVxF binding pocket on PPP1CB (red). Image generated using ChimeraX [44]. **(I)** AlphaFold3 structure of LpPIP-PPP1CB complex labeled with the active site (purple) and three substrate binding grooves (yellow) of PPP1CB.

loss of interaction with PP1 (Figs 2D and S3E). These data confirm that Lpg1625 is functioning as a PIP at the mitochondrial outer membrane for the recruitment of PP1 via the RVxF motif, which is likely the only PP1-binding motif on Lpg1625. Given this, we henceforth refer to Lpg1625 as LpPIP.

Under basal conditions, the PP1 catalytic subunits PPP1CA, PPP1CB, and PPP1CC are distributed throughout the cytosol and nucleus [46]. Their distribution is dynamically regulated by various PIPs, cellular events, and cell cycle stages [46]. Since the RVxF motif is commonly used as an anchor for PP1 recruitment, we assessed the localization of the PP1 catalytic subunit, PPP1CB in HeLa cells expressing either $^{FLAG}$LpPIP or $^{FLAG}$RVxF$_{dead}$ using immunofluorescence (Fig 2E) [47,52]. Untransfected HeLa cells showed cytosolic and nuclear localization of PPP1CB (Fig 2E, top panel). In contrast, cells expressing $^{FLAG}$LpPIP showed a proportion of PPP1CB now localized to mitochondria (Fig 2E, middle panel), whereas PPP1CB localization remained cytosolic/nuclear in cells expressing $^{FLAG}$RVxF$_{dead}$ (Fig 2E, bottom panel). This supports a critical role for the Lpg1625 RVxF motif in PP1 recruitment to mitochondria. Pearson's correlation analysis quantitatively demonstrated a significant positive colocalization between $^{FLAG}$LpPIP and PPP1CB, whereas $^{FLAG}$RVxF$_{dead}$ showed no detectable colocalization (Fig 2F). Using Fiji quantification, we calculated the proportion of PPP1CB at mitochondria in HeLa cells expressing $^{FLAG}$LpPIP, and found that approximately 17% of the protein is at mitochondria (Fig 2G).

To gain structural insight into the interaction, AlphaFold 3 was used to predict the structure of the LpPIP-PPP1CB holoenzyme [53]. The high-quality prediction revealed the RVxF motif of LpPIP is in contact with the known RVxF binding pocket on PPP1CB (Fig 2H) [47,52]. The model suggests that the PPP1CB active site, which is located at the intersection of the potential substrate binding grooves, remains accessible in the complex (Fig 2I) [54]. This suggests that PP1 retains its catalytic activity when bound to LpPIP, although potential allosteric effects cannot be excluded. Collectively, these findings show that a Legionella-specific PIP effector, LpPIP, is localized to the mitochondrial outer membrane to recruit PP1 to the organelle via its RVxF motif.

PLOS Biology

## Recruitment of PP1 to mitochondria promotes dephosphorylation of resident mitochondrial proteins

To explore the function of PP1 at mitochondria following recruitment by *Lp*PIP, we compared the mitochondrial phosphoproteomes of wild-type HEK293 cells to stable cell lines expressing either [FLAG]*Lp*PIP or [FLAG]RVxF[dead] (Fig 3A). The steady-state proteomes of these cell lines remained unchanged, eliminating the need for normalization of phospho-peptide abundance based on protein levels (Fig 3B and S5 Table). A total of 8031 proteins were detected, of which 982 were mitochondrial proteins, representing ~80% coverage of the mitochondrial proteome (Fig 3B and S5 Table) [55]. Phosphoproteomic analysis detected 18,420 peptides, of which 15,151 were phosphopeptides, corresponding to an 82% phosphopeptides enrichment. Four hundred twelve of these were phosphopeptides belonged to known mito-chondrial proteins. Cells expressing [FLAG]*Lp*PIP displayed a significant decrease in the abundance of phosphopeptides derived from mitochondrial outer membrane proteins (Fig 3C and S5 Table). This trend was consistently observed across experiments using wild-type and [FLAG]RVxF[dead] expressing cells as controls (Fig 3D). These results demonstrate that *Lp*PIP-mediated recruitment of PP1 to mitochondria leads to the dephosphorylation of specific mitochondrial outer membrane proteins, indicating a targeted mechanism by which *L. pneumophila* manipulates the host mitochondrial phosphoproteome.

To confirm the dephosphorylated proteins are specific targets of PP1 activity, seven residues flanking the target phos-phoserine/phosphothreonine (pSer/pThr) residue were aligned for motif assessment using pLogo [56]. A high frequency (35.29%) of arginine at position −3 relative to the pSer/pThr (Fig 3E) aligns with the intrinsic affinity of the PP1 catalytic subunit to dephosphorylate serine with N-terminal basic residues, in particular arginine at position −3 (RXXpS motif) [58]. This supports the hypothesis that the mitochondrial outer membrane proteins dephosphorylated by PP1 are targeted spe-cifically due to the activity of PP1 [58]. Gene ontology (GO) enrichment analysis of the dephosphorylated mitochondrial peptides/proteins revealed enrichment in processes related to mitochondrial organization and morphology and protein bio-genesis (Fig 3F) [57], which align with the functions of mitochondrial outer membrane proteins and with pathways known to be targeted by *L. pneumophila* during infection [16].

## Consequences on mitochondrial function of *Lp*PIP-PP1 dephosphorylation events

We compared the mitochondrial proteins dephosphorylated in our datasets (Fig 3D and S5 Table) with an existing phosphoproteomics dataset from HEK293 FcγR cells infected with *L. pneumophila* wild-type or a T4SS-deficient strain [17]. Notably, the phosphosites FUNDC2_S151, TOMM70_S91_S110, and MFF_S155_S157, which were among the top dephosphorylated sites in the presence of *Lp*PIP (Fig 3D and S5 Table), were significantly dephosphorylated in HEK293 FcγR cells infected with wild-type *L. pneumophila* for 8 h compared to T4SS-deficient strain [17]. This overlap suggests that *Lp*PIP contributes to the dephosphorylation of these mitochondrial phosphosites during *L. pneumophila* infection. We wanted to explore the potential functional implications on mitochondrial morphology, protein biogenesis, and mitochondrial health, given the readout of our phosphoproteomics.

First, we assessed whether cellular bioenergetics (a measure of mitochondrial health) was affected by *Lp*PIP expres-sion. We measured the oxygen consumption rate (mitochondrial oxidative phosphorylation) and extracellular acidification rate (glycolysis) in wild-type and stable cell lines expressing [FLAG]*Lp*PIP or [FLAG]RVxF[dead], but did not observe any significant difference (Fig 4A and 4B). FUN14 domain containing 2 (FUNDC2) is an outer membrane protein implicated in various biological functions, including apoptosis signaling and regulation of mitochondrial dynamics through inhibition of Mito-fusin 1 (Mfn1) [59,60]. The functional implications of FUNDC2_pS151 dephosphorylation have not been determined [60]. FUNDC2 has been reported to inhibit caspase-3 cleavage and apoptosis [59] and ectopic expression of [GFP]*Lp*PIP has been shown to induce caspase-3 cleavage [39]. In line with this, stable cell lines expressing [FLAG]*Lp*PIP showed increased caspase-3 cleavage upon staurosporine treatment compared to wild-type cells; however, this effect was independent of PP1 recruitment as it was also observed with [FLAG]RVxF[dead] expression (Fig 4C).

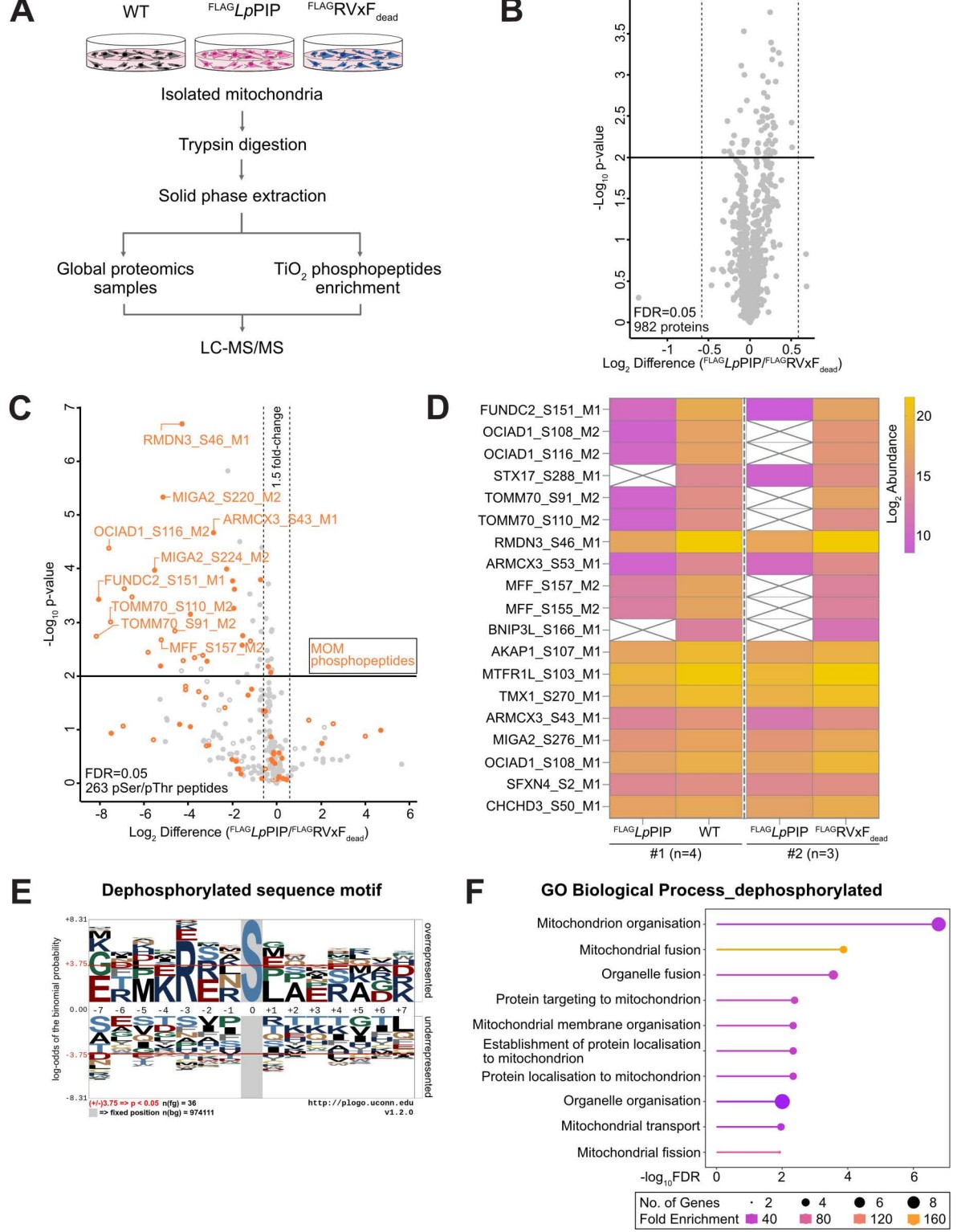

**Fig 3. Recruitment of PP1 by *Lp*PIP dephosphorylates mitochondrial proteins. (A)** Schematic of the phosphoproteomics workflow used in this study. Wild-type Flp-In T-REx 293 cells and stable cell lines expressing FLAG*Lp*PIP or FLAGRVxF_dead were induced with 1 μg/mL tetracycline for 24 h. Proteins from isolated mitochondria were trypsinized and cleaned by solid phase extraction before phosphopeptide enrichment using titanium dioxide (TiO$_2$) beads. Global

proteomics and phosphopeptide-enriched samples were analyzed by quantitative mass spectrometry and processed using Spectronaut and Perseus. **(B)** Scatterplot comparing mitochondrial protein abundance between Flp-In T-REx 293 stable cells expressing FLAG*Lp*PIP or FLAGRVxF_dead ($n=3$ technical replicates). Log$_2$ fold change of mean LFQ intensity is plotted against −Log$_{10}$ $p$-value. Significance thresholds were set at ± 1.5-fold change and $p$-value=0.01. Corresponding data are available in the Supporting information (S5 Table). **(C)** Scatterplot comparing mitochondrial phosphopeptide abundances between Flp-In T-REx 293 cells expressing FLAG*Lp*PIP and FLAGRVxF_dead ($n=3$ technical replicates). Data was filtered for mitochondrial and mitochondrial-interacting proteins using the MitoCarta3.0 annotation and restricted to serine/threonine phosphopeptides. Significance thresholds were set at ± 1.5-fold change and $p$-value of 0.01 using Student $t$ test. Phosphopeptides from mitochondrial outer membrane proteins are labeled in orange. Hollow circles indicate phosphopeptides with imputed values; solid circles represent those without. Corresponding data are available in the Supporting information (S5 Table). **(D)** Heatmap of the mitochondrial phosphopeptides significantly reduced in Flp-In T-REx 293 stable cells expressing FLAG*Lp*PIP relative to wild-type ($n=4$ technical replicates) and cells expressing FLAGRVxF_dead ($n=3$ technical replicates). Average of the Log$_2$ abundance of phosphopeptide for each group was used to generate the heatmap in Prism 10. Missing values are indicated by crosses. Corresponding data are available in the Supporting information (S5 Table). **(E)** pLogo [56] sequence motif analysis of the seven residues flanking 39 phosphoserine/phosphothreonine (pSer/pThr) sites significantly dephosphorylated in FLAG*Lp*PIP expressing cells (Fig 3C). Arginine (R) at position −3 relative to pSer/pThr was significantly enriched (log-odd probability of 6.213 and $p$-value of 1.715 e−4). **(F)** Gene ontology (GO) biological process enrichment analysis of mitochondrial proteins with significantly downregulated phosphopeptides (Fig 3D), performed using ShinyGO 0.80 [57]. The top 10 enriched GO terms are shown. Corresponding data are available in the Supporting information (S2 Data).

Mitochondrial fission factor (MFF) functions to recruit the mitochondrial fission protein Drp1 [61]. Given this and the reported function of FUNDC2 in mitochondrial morphology, we quantified organelle morphology in FLAG*Lp*PIP or FLAGRVxF-dead expressing cells, but found no changes to tubule length or network complexity (Fig 4D and 4E). Phosphorylation of MFF S155 and S172 promotes Drp1 recruitment to mitochondria, leading to mitochondrial fragmentation [62]. In the presence of FLAG*Lp*PIP, S155 and S157 of MFF were dephosphorylated (Fig 3D), but the basal proportion of Drp1 at mitochondria was not influenced (Fig 4F and 4G).

TOMM70 is an import receptor of the outer membrane TOM complex [63] that recognises carrier proteins as part of the TIM22 pathway for the import of these hydrophobic substrates into the inner membrane [63,64]. Phosphorylation of S91 on TOMM70 has been shown to enhance the import of mitochondrial carrier proteins [63]. Thus, S91 and S110 dephosphorylated in the presence of FLAG*Lp*PIP, could impair import of TIM22 substrates. However, in vitro import and assembly of the TIM22 substrates, [35S]-Tim23 and [35S]-GC1, was comparable across mitochondria isolated from wild-type and stable cell lines expressing FLAG*Lp*PIP or FLAGRVxF_dead (Fig 4H and 4I). We also assessed BNIP3L, which contains a LC3-interacting region that facilitates autophagosome recruitment to mitochondria [65], and was dephosphorylated at S166 (Fig 3D). Previous studies have shown that both PP1 and PP2A can dephosphorylate the mitophagy receptor, BNIP3, leading to its degradation and suppression of mitophagy [66]. mt-Keima, a mitochondrial-targeted, pH-sensitive fluorescent protein that enables quantification of mitochondria undergoing lysosomal degradation [67], revealed no significant differences in basal mitophagy levels due to *Lp*PIP expression (Fig 4J).

Taken together, these data suggest that *Legionella* effectors can interact with mitochondrial components in ways that may not produce immediate or easily detectable changes in cellular phenotype, possibly due to factors such as compensatory mechanisms within the host cell, or the need for specific environmental conditions, in relevant cell lineages to reveal their effects.

### *Lp*PIP-PP1 targets substrates based on cellular location

As a final question, we wanted to investigate if *Lp*PIP has any role in determining PP1 substrate specificity at the mitochondrial outer membrane. To do this, we redirected *Lp*PIP-PP1 to the ER and examined the dephosphorylation substrate profile when *Lp*PIP was no longer mitochondrially-localized. Although the K128A and K97A/R100A/K128A charge mutants of *Lp*PIP partially localized to the ER, they still colocalized with mitochondria (S2E and S4A Figs). To circumvent this, we generated a chimera consisting of the cytosolic domain of *Lp*PIP (amino acids 1–103) fused to the tail-anchor region (amino acids 109–134) of the ER TA protein CYB5A (Fig 5A) [41]. The resulting construct, FLAG*Lp*PIP-CYB5A^TA, colocalized with the ER marker PDI in transfected HeLa cells (Figs 5A and S4A). FLAG*Lp*PIP-CYB5A^TA recruited PPP1CB to the ER in transfected HeLa cells, as shown quantitatively with the Pearson's correlation analysis (Figs 5B and S4B).

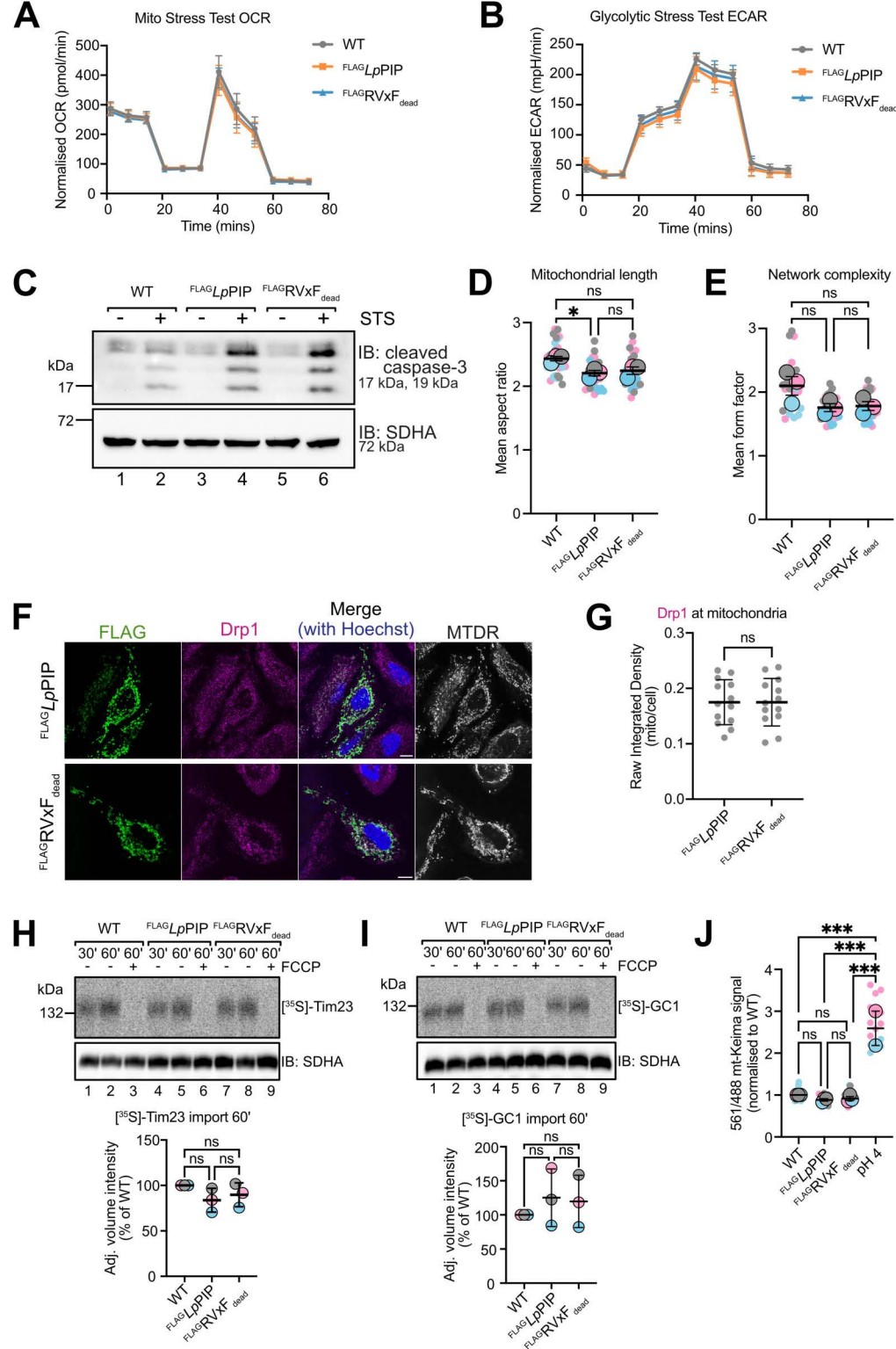

**Fig 4. Downstream consequences of *Lp*PIP-PP1 dephosphorylation events. (A)** Time course trace of oxygen consumption rate (OCR) in wild-type Flp-In T-REx 293 cells (16 technical replicates) and stable cells expressing ᶠᴸᴬᴳ*Lp*PIP (15 technical replicates) or ᶠᴸᴬᴳRVxF_dead (15 technical replicates), following injections of oligomycin, FCCP, rotenone, and antimycin A. Data represents mean ± SD. Corresponding raw data are available in the Supporting

Information (S3 Data). **(B)** Time course trace of extracellular acidification rate (ECAR) for the cells described in Fig 4A, following incubation with oligo-mycin and 2-deoxy-D-glucose. Data represents mean ± SD. Corresponding raw data are available in the Supporting Information (S3 Data). **(C)** Wild-type Flp-In T-REx 293 cells and stable cells expressing $^{FLAG}Lp$PIP or $^{FLAG}$RVxF$_{dead}$ were induced with 1 μg/mL tetracycline for 24 h, followed by treatment with 1 μM staurosporine (STS) for 24 h. Whole cell samples were processed for SDS-PAGE and immunoblots. Corresponding raw images are available in the Supporting Information (S1 Raw Images). **(D)** Quantification of mitochondrial mean aspect ratio in wild-type HeLa cells and those transfected with $^{FLAG}Lp$PIP or $^{FLAG}$RVxF$_{dead}$ ($n = 3$ independent experiments, 10 cells per experiment). Data are presented as mean ± SEM of the three experiments. Ordi-nary one-way ANOVA was performed on the means, showing a significant difference of 0.2273 between wild-type and $^{FLAG}Lp$PIP-transfected cells, with a $p$-value of 0.0309. Corresponding raw data are available in the Supporting information (S1 Data). **(E)** Quantification of mitochondrial mean form factor in HeLa cells and those transfected with $^{FLAG}Lp$PIP or $^{FLAG}$RVxF$_{dead}$ ($n = 3$ independent experiments, 10 cells in each experiment). Data are presented as mean ± SEM of the three experiments. Ordinary one-way ANOVA was performed on the means, showing no significant differences among the groups. Corresponding raw data are available in the Supporting information (S1 Data). **(F)** HeLa cells transiently transfected with $^{FLAG}Lp$PIP, or $^{FLAG}$RVxF$_{dead}$ were stained with 100 nM MitoTracker Deep Red FM (mitochondria; MTDR) and immunostained with antibodies against FLAG (green) and Drp1 (magenta). Nuclei were stained with Hoechst 33258 (blue). Scale bar represents 10 μm. **(G)** Proportion of Drp1 localized to mitochondria in HeLa cells transiently transfected with $^{FLAG}Lp$PIP or $^{FLAG}$RVxF$_{dead}$, calculated using raw integrated density in Fiji. Representative images are shown in Fig 3F. Data are pre-sented as mean ± SD from 13 cells in one experiment. Unpaired $t$ test showed no significant difference between the groups. Corresponding raw data are available in the Supporting information (S1 Data). **(H)** [$^{35}$S]-Tim23 was imported into mitochondria isolated from wild-type Flp-In T-REx 293 cells and stable cells expressing $^{FLAG}Lp$PIP or $^{FLAG}$RVxF$_{dead}$. Following import mitochondria were isolated and treated with PK. Isolated mitochondria were solubilized in 1% [w/v] digitonin and the separated by BN-PAGE and observed by phosphor image analysis. Densitometric quantification was calculated as the percentage of import at 60 min in mitochondria from wild-type Flp-In T-REx 293 cells, normalized to SDHA loading control. Data are presented as mean ± SD of three experiments. Ordinary one-way ANOVA showed no significant difference among the three cell lines. Corresponding raw data are available in the Supporting Information (S1 Raw Images and S4 Data). **(I)** [$^{35}$S]-GC1 was imported into mitochondria isolated from wild-type Flp-In T-REx 293 cells and stable cells expressing $^{FLAG}Lp$PIP or $^{FLAG}$RVxF$_{dead}$ as described in Fig 4H. Densitometric quantification and ordinary one-way ANOVA showed no significant difference among the three cell lines. Corresponding raw data are available in the Supporting information (S1 Raw Images and S4 Data). **(J)** Quantification of mitophagy from live-cell imaging of Flp-In T-REx 293 cell lines constitutively expressing mt-Keima. Experimental conditions include wild-type Flp-In T-REx 293 cells cultured in normal media or media adjusted to pH 4 (positive control), and stable cells induced to express either $^{FLAG}Lp$PIP or $^{FLAG}$RVxF$_{dead}$. The 561/488 mt-Keima signal (red-to-green fluorescence ratio) was quantified using the Ratio Plus plugin in Fiji and normal-ized to that of wild-type cells in normal media. Data represent mean ± SEM from two to three independent experiments (>30 images per experiment). Statistical analysis by ordinary one-way ANOVA showed no significant differences among the cell lines, except for the pH 4 control. Corresponding raw data are available in the Supporting information (S1 Data).

Co-immunoprecipitation of $^{FLAG}Lp$PIP or $^{FLAG}Lp$PIP-CYB5A$^{TA}$ from whole cell lysates of transfected HeLa cells revealed that the ER-localized $Lp$PIP chimera interacted with fewer mitochondrial proteins and more ER proteins and it continued to robustly interact with PP1 (Fig 5C and 5D and S6 Table). Whole cell phosphoproteomics (S7 Table) to determine the substrates of the ER-localized $Lp$PIP-CYB5A$^{TA}$-PP1 revealed an increase in ER proteins and reduction in mitochondrial proteins being dephosphorylated in cells expressing $^{FLAG}Lp$PIP-CYB5A$^{TA}$ (Figs 5E, 5F, S4C and S4D and S7 Table). These data suggested that PP1's substrate specificity is largely determined by its localization and its proximity to organelle-specific proteins. Sequestering PP1 to either mitochondria or ER, thereby reducing its abundance from the typical cytoso-lic and nuclear locations, is expected to broadly disrupt the cellular phosphorylation landscape. This effect was observed in cells overexpressing $^{FLAG}Lp$PIP or $^{FLAG}Lp$PIP-CYB5A$^{TA}$, which showed increased phosphorylation of proteins predomi-nantly localized to the nucleus (S4G and S4H Fig).

## Discussion

Phosphorylation is a prevalent post-translational modification in eukaryotic cells, acting as a key regulatory mechanism that governs a wide range of cellular functions. The host phosphoproteome is therefore a critical target during infection, and *L. pneumophila* has evolved numerous strategies to manipulate it through its effector arsenal [17]. Several *L. pneu-mophila* T4SS effectors have been reported to either induce phosphorylation or dephosphorylation of host proteins [68]. For example, *L. pneumophila* protein kinases such as LegK1, LegK2, LegK3, LegK4, and LegK7 phosphorylate host proteins to prevent lysosomal degradation of *Legionella*-containing vacuoles, inhibit host programmed cell death, inhibit host protein synthesis, and modulate transcription [69–73]. Conversely, effectors with tyrosine phosphatase activity (Ceg4, Lem4, and WipA) inhibit mitogen-activated protein kinase signaling and disrupt host actin polymerization [74–76]. Additionally, WipB, containing a serine/threonine phosphatase domain, regulates host lysosomal nutrient sensing [77].

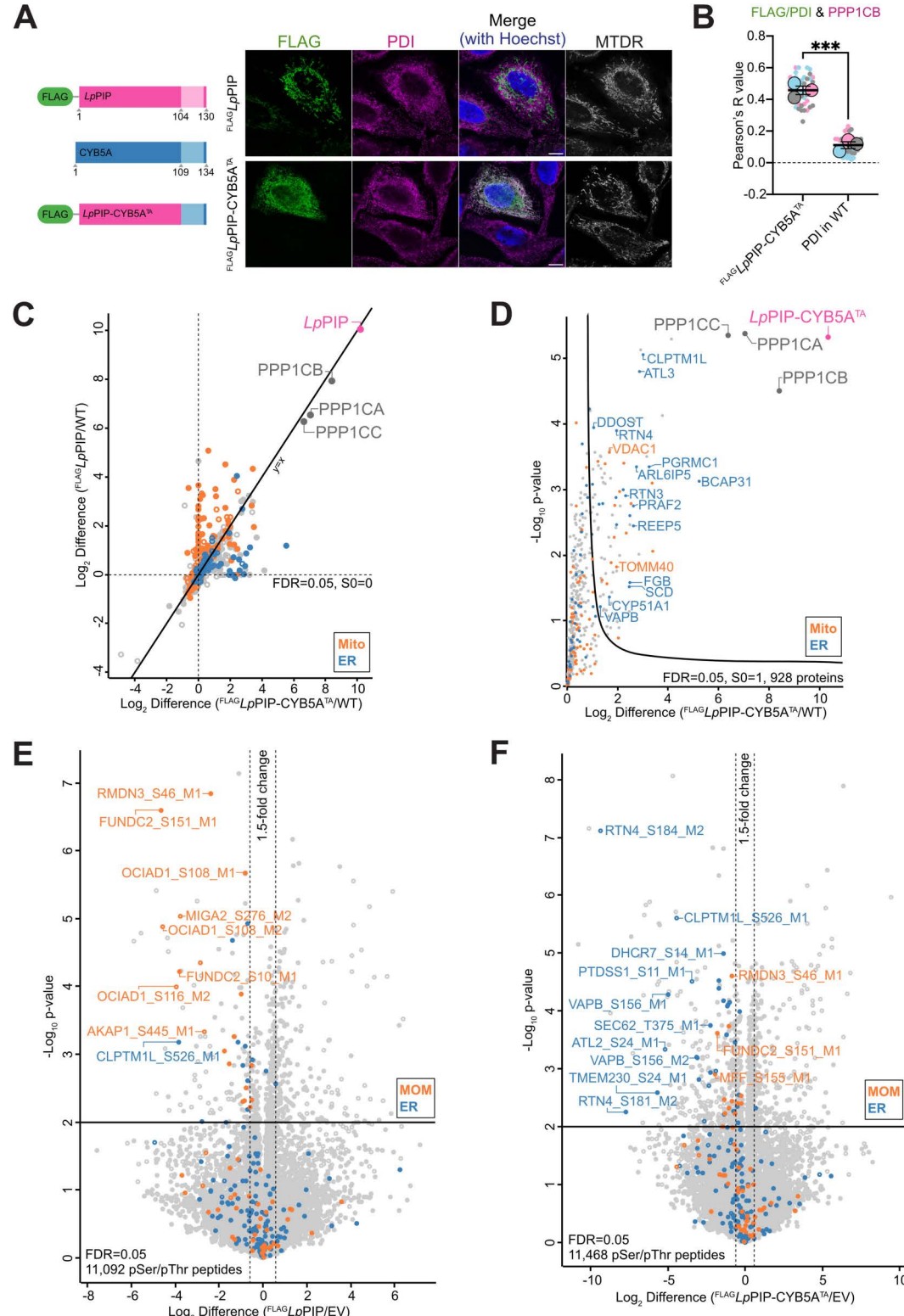

**Fig 5. LpPIP does not specify substrates selectivity for PP1. (A)** HeLa cells transiently transfected with $^{FLAG}$LpPIP or $^{FLAG}$LpPIP-CYB5A$^{TA}$ were stained with 100 nM MitoTracker Deep Red FM (mitochondria; MTDR) and immunostained with antibodies against FLAG (green) and PDI (ER;

magenta). Nuclei were stained with Hoechst 33258 (blue). Scale bar represents 10 µm. **(B)** Pearson correlation analysis of FLAG or PDI and PPP1CB fluorescence signals was performed on $^{FLAG}Lp$PIP-CYB5A$^{TA}$ expressing HeLa cells and untransfected wild-type HeLa cells ($n$ = 3 independent experiments, 16 cells per experiment). Data are shown as the mean ± SEM of three independent experiments. Larger circles represent mean values from each experiment, and smaller circles represent individual cell values. Unpaired $t$ test was performed on the means, revealing a significant difference of 0.35 ± 0.03 between the two groups, with a $p$-value of 0.0005. Representative images are shown in S4B Fig. Corresponding raw data are available in the Supporting information (S1 Data). **(C)** Whole cell FLAG co-immunoprecipitation was performed using untransfected HeLa cells, HeLa cells expressing $^{FLAG}Lp$PIP, or $^{FLAG}Lp$PIP-CYB5A$^{TA}$ ($n$ = 3 technical replicates). The Log$_2$ difference of proteins enriched in $^{FLAG}Lp$PIP and $^{FLAG}Lp$PIP-CYB5A$^{TA}$ relative to untransfected cells is shown. Proteins with significant fold change (± 1.5-fold change with a $p$-value ≤ 0.05) are represented as filled circles, while non-significant differences are shown as hollow circles. Mitochondrial proteins based on MitoCarta3.0 are indicated in orange, and ER proteins based on the Human Protein Atlas are indicated in blue. Corresponding data are available in the Supporting information (S6 Table). **(D)** Volcano plot showing proteins enriched following FLAG co-immunoprecipitation from HeLa cells expressing $^{FLAG}Lp$PIP-CYB5A$^{TA}$ compared to untransfected cells ($n$ = 3 technical replicates). Mitochondrial proteins based on MitoCarta3.0 are indicated in orange, and ER proteins based on the Human Protein Atlas are indicated in blue. Corresponding data are available in the Supporting information (S6 Table). **(E)** Scatterplot comparing phosphopeptides abundances in whole cell samples from HeLa cells transfected with $^{FLAG}Lp$PIP to empty vector pCDNA5 ($n$ = 4 technical replicates). Data were filtered for serine and threonine phosphopeptides. The significance thresholds were set to a ±1.5-fold change with a $p$-value of 0.01 using Student $t$ test. Phosphopeptides derived from mitochondrial outer membrane proteins (MitoCarta3.0) and ER proteins (Human Protein Atlas) are labeled in orange and blue, respectively. Non-imputed phosphopeptides are shown as filled circles, while phosphopeptides with imputed values are shown as hollow circles. Corresponding data are available in the Supporting information (S7 Table). **(F)** Scatterplot comparing phosphopeptides abundances in whole cell samples from HeLa cells transfected with $^{FLAG}Lp$PIP-CYB5A$^{TA}$ to empty vector pCDNA5 ($n$ = 4 technical replicates), processed as described in Fig 5E. Corresponding data are available in the Supporting information (S7 Table).

However, this is the first report of *L. pneumophila* modulating the mitochondrial phosphoproteome through the action of an effector protein.

Rather than acting directly as a kinase or phosphatase, our study shows that $Lp$PIP functions as a PIP, redirecting host PP1 to the mitochondria, where it dephosphorylates mitochondrial outer membrane proteins. Notably, expression of $Lp$PIP is significantly upregulated during *L. pneumophila* intracellular replication in macrophages [78], suggesting that its function contributes to the intracellular success of this pathogen. Given the broad substrate specificity of PP1, it is involved in diverse cellular processes, including the cell cycle, protein synthesis, apoptosis, glycogen metabolism, and actin cytoskeleton organization [79]. By sequestering PP1 to mitochondria and away from its typical cytosolic and nuclear localization, $Lp$PIP-PP1 could perturb the host phosphorylation landscape far beyond the mitochondrial compartment, as supported by our ectopic expression models (S4G and S4H Fig). Indeed, since we did not observe any significant functional consequences at the mitochondria, it raises the possibility that the bacterium may deliberately sequester PP1 to this organelle to divert the phosphatase away from the nucleus or cytosol. Although we did not have the opportunity to explore this possibility, it remains a compelling hypothesis that warrants consideration.

Mitochondrial phosphorylation has emerged as a critical regulatory mechanism despite being maintained at low stoichiometry, likely due to the presence of mitochondrial-localized phosphatases [80]. Although most mitochondrial phosphorylation sites remain functionally uncharacterized, several studies underscore their importance. For example, deletion of the mitochondrial matrix phosphatase PPTC7 leads to increased mitophagy, highlighting the role of mitochondrial phosphoregulation in mitochondrial quality control [81,82]. The mitochondrial phosphatase PGAM5 can localize to the mitochondrial outer membrane, where it dephosphorylates MFN2 to promote mitochondrial fusion [83]. Additionally, the phosphorylation state of mitochondrial precursors can affect mitochondrial import, thereby regulating mitochondrial biogenesis [84]. Interestingly, PP1 activity has previously been observed at mitochondria, where it dephosphorylates the anti-apoptotic protein Bcl-2 during late-mitosis [85,86], supporting a role of PP1 in regulating mitochondrial function. The dephosphorylation of outer membrane proteins by $Lp$PIP-PP1 has confirmed previously published phosphorylation sites that have not been functionally characterized. This highlights the power of using *Legionella* as a model to uncover novel aspects of eukaryotic cell biology.

The downstream consequences of the targeted dephosphorylation of mitochondrial outer membrane proteins identified in our study remain elusive. It is plausible that the presence of *L. pneumophila* infection may be necessary to reveal relevant phenotypes, if *Lp*PIP-PP1 targets other effectors localized to the mitochondrial outer membrane. The functional impact of *Lp*PIP-PP1 activity may be shaped by interplay with additional effectors. For instance, the T4SS effector LnaB catalyzes AMPylation of phosphorylated residues, adding an AMP moiety to the phosphate chain to generate a novel post-translational modification termed ADPylation [87]. This additional layer of regulation could influence the susceptibility of modified substrates to PP1-mediated dephosphorylation. As mentioned above, *Lp*PIP-mediated PP1 relocalization may exert broader cellular effects by altering phosphorylation of cytosolic or nuclear proteins.

Given that over 200 mammalian PIPs were validated to regulate PP1 activity, it is reasonable to speculate that additional *L. pneumophila* effectors could function similarly to *Lp*PIP to control host PP1 function. Indeed, we identified 11 *L. pneumophila* T4SS effectors that encode a RVxF motif, including LegK4 and LegK7 (S2 Text), which raises the intriguing possibility that these effectors could interact with PP1 and potentially influence its activity in a spatiotemporal manner. Notably, PP1 can be inhibited via phosphorylation by kinases [88], suggesting that effectors with dual kinase and PIP activity (potentially LegK4 and LegK7) might fine-tune phosphatase activity to amplify substrate phosphorylation.

In summary, we identified a novel *L. pneumophila* effector, *Lp*PIP, that hijacks host PP1 to modulate mitochondrial phosphoproteome. This work expands our understanding of bacterial subversion of host cell signaling and highlights a unique strategy whereby a pathogen effector relocalizes a host phosphatase to control cellular function. A key limitation of our study is that these findings were derived from ectopic expression systems in the absence of infection. Given that *L. pneumophila* translocates hundreds of effectors and triggers extensive host cell reprogramming during infection, the function, localization, or substrate specificity of *Lp*PIP may differ in that context. Future work will be critical to validate these observations under infection conditions and fully elucidate the physiological relevance of *Lp*PIP-mediated phosphoregulation. In addition to *Lp*PIP, our study uncovered additional mitochondrial TA effectors and other RVxF-containing effectors. These effectors may offer further insights into how *L. pneumophila* subverts the host phosphorylation landscape and modulates the dynamic interface between mitochondria and cytosol.

## Materials and methods

### Cell culture and cell lines

Flp-In T-REx 293 (ThermoFisher Scientific #R75007; RRID: CVCL_U427) and HeLa cells (RRID: CVCL_0030) were cultured in Dulbecco's modified Eagle's medium (DMEM; Gibco # 11995073) supplemented with 5% [v/v] heat-inactivated fetal bovine serum (FCS; Gibco #A3382001) and 1% [v/v] penicillin-streptomycin (pen/strep; Gibco #15-140-122). THP-1 cells (ATCC # TIB-202) were cultured in Roswell Park Memorial Institute 1640 (RPMI) medium (Gibco, # 61870127) supplemented with 10% [v/v] heat-inactivated FCS (Bovagen # FBSAU-2207C). All were incubated at 37 °C with 5% $CO_2$ in a humidified incubator. $^{FLAG}$Lpg1625 and $^{FLAG}$Lpg1625$^{K17A/V19A}$ stable cell lines were all generated from Flp-In T-REx 293 cells.

### Bacterial cell cultures and strains

*Escherichia coli* strains DH5α and Pir2 were used for the propagation of genetic material and were cultured at 37 °C on Luria–Bertani (LB) agar or broth supplemented with 50 µg/mL ampicillin, 25 µg/mL chloramphenicol, or 15 µg/mL kanamycin where required. *Legionella pneumophila* Philadelphia-1 strain JR32 was grown on Buffered-Charcoal Yeast Extract (BCYE) agar supplemented with 135 µg/mL iron (III) nitrate nonahydrate and 400 µg/mL L-cysteine at 37 °C. Where appropriate, the bacteria were selected using 6 µg/mL chloramphenicol, 15 µg/mL kanamycin, or 100 µg/mL streptomycin. *L. pneumophila* was transformed via triparental mating [89] or electroporation [90]. *E. coli* strain HB101 carrying a pRK600 mobilization plasmid was used in triparental mating to facilitate the uptake of the donor DNA by the *L. pneumophila* recipients.

## Molecular biology

*L. pneumophila* gene sequence was obtained from NCBI. Genomic DNA isolated from wild-type *L. pneumophila* Philadelphia-1 strain JR32 was used as template for the amplification of *lpg1625, lpg1803, lpg2444*, and *lpg2344*. These were cloned into the pCDNA5 vector with a FLAG tag encoded upstream. Site-directed mutagenesis was performed using Q5 site-directed mutagenesis kit (NEB #E0552S). Oligonucleotides were designed with the online NEB primer design software (NEBaseChanger.neb.com).

## Genetic manipulation of *L. pneumophila*

The Δ*lpg1625* strain was generated using an established method [90]. Briefly, the up- and downstream regions flanking the *lpg1625* open reading frame were amplified from genomic DNA and ligated by overlap extension PCR. The resulting DNA fragment was inserted into the R6K vector pSR47s and introduced into *L. pneumophila* by triparental mating. The vector was integrated into the genome by homologous recombination and the target gene was subsequently removed upon resolution of pSR47s by sucrose counter-selection. Knock-out (KO) clones were identified using MyTaq HS Red polymerase as per the manufacturer's instructions, using primers screening across the region of interest and primers targeting *lpg1625*.

The Δ*lpg1625* strain was complemented with constitutively expressed gene variants. The previously generated constructs, $^{FLAG}$Lpg1625 and $^{FLAG}$RVxF$_{dead}$, were inserted into a non-integrative pMMB207 plasmid and introduced into *L. pneumophila* Δ*lpg1625* by electroporation. Expression of Lpg1625 derivatives was induced with 1 mM isopropyl ß-D-1-thiogalactopyranoside (IPTG), which was validated by western blot analysis.

## Transfection and stable cell lines generation

Lipofectamine 3000 (ThermoFisher Scientific # L3000008) was used for transfection as per the manufacturer's protocol. Briefly, cells were seeded to be approximately 70% confluent on the day of transfection. For generation of tetracycline-inducible stable cell lines, Flp-In T-REx 293 cells were co-transfected with pOG44 and pCDNA5 plasmids encoding the gene of interest. Following 72 h, cells were placed on selection with 200 µg/mL Hygromycin B (Gibco #10687010). Selection media was replaced as required until all non-transfected control cells were dead. Cells that survived the hygromycin selection were pooled and screened for expression of the protein of interest after 24 h induction with 1 µg/mL tetracycline (Sigma-Aldrich #T7660).

For generation of stable cell lines constitutively expressing mt-Keima, retrovirus was packaged in HEK293T cells following co-transfection with 9 µg pCHAC-mt-mKeima (Addgene #72342), 6 µg pUMVC3, and 3 µg pCMV-VSVG in a 10 cm tissue culture dish. After 48 h, the viral supernatant was collected, filtered through a 0.45 µm Millex-HV syringe filter unit (Millipore), and used to transduce Flp-In T-REx 293, $^{FLAG}$Lpg1625 Flp-In T-REx 293, and $^{FLAG}$Lpg1625$^{K17A/V19A}$ Flp-In T-REx 293 cells in a 6-well plate for 24 h in the presence of 8 µg/mL polybrene (Sigma-Aldrich #107689). mt-Keima-positive cells were subsequently acquired via fluorescence-activated cell sorting.

## Infection of THP-1 cells with *L. pneumophila*

Intracellular replication of wild-type and mutant strains of *L. pneumophila* in THP-1 cells was assessed by counting colony-forming units (CFU/mL). THP-1 cells were seeded at a density of $5 \times 10^5$ cells and differentiated into macrophage-like cells with 10 nM phorbol 12-myristate 13-acetate (PMA) 72 h prior to infection. Stationary-phase *L. pneumophila* was resuspended in PBS and diluted to a multiplicity of infection (MOI) of 5 in RPMI containing 1 mM of IPTG, where appropriate. The *L. pneumophila* inoculum was incubated with THP-1 cells for 2 h at 37 °C and 5% $CO_2$ to allow bacterial internalization before the remaining extracellular bacteria were killed with 100 µg/mL gentamicin in RPMI for 1 h at 37 °C and 5% $CO_2$. At the appropriate time points post-infection, the THP-1 cells were lysed with 0.05% digitonin in PBS. The harvested samples

were serially diluted and plated onto BCYE agar and incubated for 4 days at 37 °C before quantifying CFU/mL. Five biological replicates were used for the quantification. Statistical test was carried out with Prism 10 using two-way ANOVA.

## Oxygen consumption rate and extracellular acidification rate measurement

Mitochondrial stress and glycolytic stress tests were performed using a Seahorse Bioscience XFe96 Analyzer according to manufacturer protocols. Forty-eight hours before the assay, $1.5 \times 10^4$ of wild-type Flp-In T-REx 293 (16 technical replicates), [FLAG]Lpg1625 Flp-In T-REx 293 (15 technical replicates), and [FLAG]Lpg1625[K17A/V19A] Flp-In T-REx 293 stable cells (15 technical replicates) were plated per well in XFe96 culture plates pre-coated with 100 μg/mL poly-D-Lysine and grown overnight under standard culture conditions. Twenty-four hours before the assay, cells were induced with 1 μg/mL tetracycline (Sigma-Aldrich #T7660). Cells were assayed in non-buffered DMEM media (Agilent #102353-100) containing either 10 mM glucose, 1 mM sodium pyruvate, and 2 mM glutamine (mitochondrial stress test) or 2 mM glutamine only (glycolytic stress test). For the mitochondrial stress test, rates were measured following sequential incubations with 2 μM oligomycin, 0.5 μM carbonyl cyanide 4-(trifluoromethoxy) phenylhydrazone (FCCP), and 0.5 μM rotenone and 0.3 μM antimycin A. For the glycolytic stress test, these incubations instead contained 10 mM glucose, 2 μM oligomycin, and 50 mM 2-deoxy-D-glucose. For both tests, measurements consisted of 3 cycles each of 3 min mix and 3 min measure. Following the assay, cell numbers per well were normalized using CyQuant (Thermo Fisher Scientific #C7026).

## Immunofluorescence and confocal microscopy

Twenty-two mm glass coverslips were sterilized with 75% [v/v] ethanol and placed in 6-well plates. HeLa cells were seeded onto the coverslips the day before transfection. When required, cells were stained with 100 nM MitoTracker Deep Red FM (ThermoFisher Scientific #M22426) in serum-free DMEM for 40 min at 37 °C. Cells were then fixed in 4% [w/v] paraformaldehyde (PFA; ProSci Tech #C004) diluted in 5% [w/v] sucrose (Chem-Supply #SA030) for 10 min at room temperature. For ER staining, 0.1% [w/v] glutaraldehyde (Sigma-Aldrich #G7526) was included for the fixation. The cells were washed with phosphate buffered saline (PBS; 137 mM NaCl, 2.7 mM KCl, 10 mM $Na_2HPO_4$, 1.8 mM $KH_2OP_4$, pH 7.4) three times before being permeabilised by 0.1% [v/v] Triton X-100 (Sigma-Aldrich #T9284-500ML) in PBS (3× over 10 min). Samples were blocked with 3% [w/v] bovine serum albumin (BSA; Sigma-Aldrich #A7030) in PBS for 30 min, followed with three washes of PBS. Primary and secondary antibodies were diluted appropriately with 3% BSA in PBS. The coverslips were incubated with the primary antibodies for 1 h at room temperature. Coverslips were washed with PBS (4× over 10 min) before incubating with secondary antibodies for 1 h at room temperature. Three washes with PBS followed, with approximately 0.01 μg/μL Hoechst 33258 (Sigma-Aldrich #14530) added to the second wash for DNA staining. The coverslips were then mounted to microscope slides using approximately 50 μL of mounting medium (0.2 M 1,4-diazabicyclo[2.2.2]octane (Sigma-Aldrich #D27802), 80 mM Tris-Cl pH 8.0, 90% [v/v] glycerol) and sealed with nail polish. The slides were imaged using a Leica SP8 confocal microscope and processed with Fiji software in ImageJ.

Primary antibodies used for immunofluorescence included: anti-FLAG (Sigma #F1804, dilution 1:500), anti-NDUFAF2 (Ryan Lab, Monash University, made in-house, dilution 1:500), anti-PPP1CB (Abcam #ab53315, dilution 1:500), anti-PDI (Enzo # ADI-SPA-891-D, dilution 1:300), anti-PEX14 (Proteintech #10594-1-AP, dilution 1:500), anti-GM130 (Abcam #ab52649, dilution 1:600; gifted from Gleeson Lab, University of Melbourne), and anti-Drp1 (BD Biosciences #611113, dilution 1:500). Secondary antibodies used were AlexaFluor488 (Invitrogen #A11001, dilution 1:500), and AlexaFluor568 (Invitrogen #A11011, dilution 1:500).

## Live-cell imaging for mt-Keima

Prior to live-cell imaging, cells were seeded onto FluoroDish cell culture dishes (World Precision Instruments) and grown for 1 day (37 °C, 5% $CO_2$) in the presence of 1 μg/mL tetracycline for 24 h to induce protein expression. The following day, confocal imaging was conducted using a Zeiss Elyra LSM880 (Carl Zeiss MicroImaging) equipped with a

Plan-Apochromat 63×/1.40-NA oil immersion objective lens. mt-Keima fluorescence was imaged by sequential excitation with Argon 25 mW 458 and 561 nm 20 mW lasers, and emission was detected with BP 495-550 + LP570 filter. Control pH 4 samples were prepared as described previously [91]. Briefly, cells were permeabilised with 0.01% Triton X-100 in PBS and equilibrated in DMEM adjusted to pH 4.0. Imaging parameters, including laser power and gain, were kept consistent across all samples and conditions. Images were analyzed using the Ratio Plus plugin for Fiji image software [92] to determine the 561/488 mt-Keima ratio, which was normalized to the 561/488 ratio in wild-type cells.

### Caspase-3 activation assay

$3 \times 10^5$ wild-type Flp-In T-REx 293, FLAGLpg1625 Flp-In T-REx 293, and FLAGLpg1625K17A/V19A Flp-In T-REx 293 stable cells were plated in a 6-well plate and grown overnight under standard culture conditions. Cells were induced with 1 µg/mL tetracycline (Sigma-Aldrich #T7660) for 24 h (37 °C, 5% $CO_2$) before treating with 1 µM staurosporine (Sigma-Aldrich #GA8507) or an equivalent volume of DMSO (vehicle control) and grown for a further 24 h. Whole cell lysates were harvested for analysis by SDS-PAGE.

### Image analyses using Fiji

For fluorescence profile, a line was first drawn across a region of interest (ROI) using ROI Manager. The intensities of pixels along the line for each channel were obtained through the Plot Profile in Fiji. The data points from different channels were copied and a graph generated in Prism 10. Colocalization analysis was performed with the plugin Coloc2 [93] using Costes threshold regression. Pearson's R value (per cell) was quantified (number of cells indicated in figure legends for Figs 2F, 5B, S2B-D, and S4A) analyzed per experiment. Mean Pearson's *R* values were compared using unpaired *t* test (for two groups, one independent variable) or ordinary one-way ANOVA (more than two groups, one independent variable) or ordinary two-way ANOVA (more than two groups, two independent variables. Statistical analysis was performed using Prism 10.

To quantify the proportion of PPP1CB or Drp1 signal at mitochondria, whole cell and mitochondria ROIs were defined. Whole cell ROIs were defined manually, while mitochondrial ROIs were defined using the plugin Mitochondrial Analyzer [94]. The raw integrated density of each ROI was measured with the function Integrated density. The proportion of the signal at mitochondria was then calculated by dividing the raw integrated density at mitochondria with that from the whole cell (number of cells indicated in figure legends for Fig 2G and 4G). Statistical analysis was performed using unpaired *t* test, Prism 10.

For quantification of mitochondrial morphology, the Fiji plugin Mitochondrial Analyzer [94] was used for mitochondrial thresholding and subsequent determination of mean aspect ratio and mean form factor quantified per cell. The means of these parameters across three independent experiments were compared statistically using ordinary one-way ANOVA in Prism 10.

### Mitochondria isolation and mitochondrial treatments

Mitochondria were isolated from tissue culture cells using differential centrifugation [95]. Briefly, cells were harvested in cold PBS using a cell scraper and pelleted by centrifugation at 600*g* (5 min, 4 °C). Cells were homogenized in isolation buffer (20 mM HEPES-KOH pH 7.6, 220 mM mannitol 70 mM sucrose, 1 mM EDTA, and 0.5 mM phenylmethanesulfonyl fluoride (PMSF)) using a handheld glass Dounce homogenizer. The homogenate was centrifuged at 800*g* (10 min, 4 °C), with the resultant supernatant further centrifuged at 10,000*g* (10 mins, 4 °C). The pellets containing crude mitochondria were resuspended in isolation buffer (without PMSF). Protein concentration of the suspension was estimated using Pierce BCA protein assay kits (ThermoFisher Scientific #23225) before downstream applications/treatments. For phosphoproteomics, instead of PMSF, protease inhibitor (Roche #11873580001) and PhosSTOP (Roche #4906837001) were added into the isolation buffer and were included in every steps post homogenization.

For sub-mitochondrial fractionation, mitochondrial pellets were resuspended in isolation buffer, HEPES buffer (10 mM HEPES (Chem-Supply #HEPES05-500G), pH 7.6), or 0.5% Triton X-100 at 1 mg/mL. Mitochondrial lysates were treated with 50 µg/mL Proteinase K (PK; Sigma-Aldrich #P6556) for 10 min on ice, followed by a further 5 min incubation on ice after addition of final 1 mM PMSF in all samples. For carbonate extraction, mitochondrial pellet was resuspended in freshly made 100 mM sodium carbonate at 1 mg/mL and incubated on ice for 30 min. Centrifugation at 100,000$g$ (30 min, 4 °C) then separated the pellet (membrane proteins) and the supernatant (soluble and peripheral membrane proteins) fraction. All samples were precipitated with 12.5% [w/v] trichloroacetic acid (TCA; Chem-Supply #TA030) before analysis with SDS-PAGE.

## Mitochondrial in vitro protein import assays

Isolated mitochondrial pellets were resuspended to 1 mg/mL in import buffer (250 mM sucrose, 5 mM magnesium acetate, 80 mM potassium acetate, 10 mM sodium succinate, 1mM DTT, 5 mM ATP, 20 mM HEPES-KOH pH 7.4) with or without the addition of 10 µM carbonyl cyanide 4-(trifluoromethoxy)phenylhydrazone (FCCP; Sigma-Aldrich #C2920) to dissipate the membrane potential. Samples were incubated at 37 °C for 1 min prior to the addition of radiolabeled protein and further incubation at 37 °C with radiolabeled protein to allow for its import. Import reactions were stopped by placing samples on ice. Fifty µg/mL PK was added to samples immediately following import for 10 min on ice, followed by the addition of 1 mM PMSF for 5 min on ice. The mitochondria were isolated at 12,000$g$ for 5 min at 4 °C and washed once with import buffer. Mitochondrial pellets were then analyzed using BN-PAGE. After western transfer, radioactive signal on the PVDF membrane was exposed to a storage phosphor screen in a cassette (GE Healthcare; from 1 h up to 2 weeks). The radioactivity on the storage phosphor screen was then detected using phosphor imaging with Amersham Typhoon (GE Healthcare). If required, densitometric quantification of the radioactive signals was performed on Image Lab 6.1, normalized against SDHA loading controls.

## Tricine SDS polyacrylamide gel electrophoresis and western blotting

Tris-tricine SDS-PAGE gel was used for electrophoresis of denatured samples before immunoblot analysis [95]. 16% [v/v], 10% [v/v], and 4% [v/v] acrylamide gel mixes which constitute the gels were made with acrylamide solution (49.5% [w/v] acrylamide, 3% [w/v] bis-acrylamide) and tricine gel buffer (1 M tris, 0.1% [w/v] SDS, pH 8.45), with an addition of 13% [w/v] glycerol in the 16% [v/v] acrylamide gel mix. 10%–16% acrylamide gradient gels were poured using a gradient mixer mixing 16% [v/v] and 10% [v/v] acrylamide gel mixes. The gels were polymerized with the addition of tetramethylethylenediamine (TEMED; Sigma-Aldrich #T22500) and 10% [w/v] ammonium persulfate (APS; Sigma-Aldrich #A3678) right before pouring. A 4% [v/v] acrylamide stacking gel was poured on top of the polymerized 10%–16% gradient gel. Samples resuspended in loading dye (50 mM Tris-Cl pH 6.8, 100 mM dithiothreitol (DTT), 2% [w/v] sodium dodecyl sulfate (SDS), 10% [v/v] glycerol, 0.05% [w/v] bromophenol blue) were boiled at 65 °C for 15 min before loaded onto the gel along with PageRuler Prestained Protein Ladder (ThermoFisher Scientific #26616) as the molecular weight marker. Electrophoresis of the gels was performed with anode buffer (0.2 M Tris-Cl, pH 8.9) and cathode buffer (0.1 M Tris, 0.1 M tricine, 0.1% [w/v] SDS, pH 8.45).

After electrophoresis, SDS-PAGE gels were transferred to 0.45 µm PVDF membrane (Millipore #IPVH00010) using Owl HEP Series Semidry Electroblotting System (ThermoFisher Scientific #HEP-1). After transfer, PVDF membranes were blocked with 5% [w/v] non-fat milk (Nestle #12428935) in PBST (PBS with 0.05% Tween20 (Sigma-Aldrich #P1397)) for 1 h at room temperature. Membranes were then incubated with primary antibodies at 4 °C overnight before incubated with horseradish peroxidase-conjugated secondary antibodies (Sigma-Aldrich #A9044 and #A0545, 1:5000 dilution in 5% [w/v] non-fat milk) for 2 h at room temperature. After that, the blots were developed using Clarity Western ECL Substrate (Bio-Rad #1705061) and imaged with ChemiDoc imaging system (Bio-Rad #12003153).

Primary antibodies used for western blotting include anti-FLAG (Sigma #F1804, dilution 1:1000), anti-MFN2 (Ryan Laboratory, dilution 1:250), anti-TIMM29 (Sigma-Aldrich #HPA041858, dilution 1:1000), anti-TIMM44 (Proteintech #13859-1-AP, dilution 1:500), anti-SDHA (Abcam #AB14715, dilution 1:1000), and anti-caspase-3 (Cell Signaling Technology #9662, dilution 1:500).

## Blue native polyacrylamide gel electrophoresis (BN-PAGE)

BN-PAGE was used for the separation of native protein complexes. Acrylamide gradient gels were poured with 4% [v/v] and 16% [v/v] acrylamide gel mixes made in blue native gel buffer (66.7 mM ε-amino-n-caproic acid (Sigma-Aldrich #D7754), 50 mM Bis-Tris pH 7.0). 4% [v/v] acrylamide stacking gel was poured on top of the gradient gel. Mitochondrial samples were solubilized in 1% [w/v] digitonin (Merck #300410) in solubilization buffer (20 mM Tris-Cl pH 7, 50 mM NaCl, 10% [v/v] glycerol) on ice for 30 min. Insoluble material was removed by centrifugation at 16,000g for 20 min at 4 °C. One-tenth the volume of 10× BN loading dye (5% [w/v] Coomassie brilliant blue G-250 (MP Biomedicals #04808274), 0.5 M ε-amino-n-caproic acid, 100 mM Bis-Tris pH 7.0) was added to the clarified supernatants. Samples were loaded alongside BN molecular weight marker (0.3 mg/mL thyroglobulin, 0.3 mg/mL ferritin, and 0.6 mg/mL BSA in solubilization buffer). Electrophoresis was performed at 4 °C, with BN anode buffer (50 mM Bis-Tris, pH 7.0) and BN cathode buffer (50 mM Tricine, 15 mM Bis-Tris, with 0.02% [w/v] Coomassie brilliant blue G-250 for a third of the gel before changing to cathode buffer without it).

## Co-immunoprecipitation and mass spectrometry

Isolated mitochondria or whole cell samples were solubilized in solubilization buffer (20 mM Bis-Tris pH 7, 50 mM NaCl, 10% [v/v] glycerol, 1× protease inhibitor (Roche #11873580001)) containing 1% [w/v] digitonin (Merck #300410) at 2 mg/mL (45 min, 4 °C). After centrifugation at 16,000g (20 min, 4 °C) to pellet insoluble debris, supernatant was diluted with the solubilization buffer to a final concentration of 0.1% [w/v] digitonin before addition to pre-equilibrated anti-FLAG affinity gel (Millipore #A2220) for 60 min at 4 °C. Four washes of the anti-FLAG affinity gel with solubilization buffer containing 0.1% [w/v] digitonin were performed before elution of bound proteins with 0.2 M glycine (pH 2.5). The eluates were precipitated with 100% acetone at −20 °C overnight before downstream processing for mass spectrometry.

The acetone pellets were solubilized in urea/ABC buffer (8 M urea, 50 mM ammonium bicarbonate (ABC)) with 15 min sonication at RT. Solubilized samples were reduced and alkylated using 50 mM TCEP and 500 mM CAA, with 30 min incubation at 37 °C with 500 RPM shaking. The samples were then diluted to 2 M urea with 50 mM ABC before digestion with 1 µg trypsin (37 °C overnight). Trypsinized samples were acidified to 1% [v/v] TFA before clarification (21,000g, 10 min, RT). In-house stage-tips made with 3 M Empore SDB-XC extraction disks (Sigma-Aldrich #66884-U) were activated with 100 µL ACN and washed once with 0.1% [v/v] TFA, 2% [v/v] ACN before addition of clarified samples (2,500g, RT). Stage-tips were washed once with 0.1% [v/v] TFA, 2% [v/v] ACN before eluted in 0.1% [v/v] TFA, 80% [v/v] ACN. Eluates were dried with CentriVap Benchtop Vacuum Concentrator (Labconco) and reconstituted in 0.1% [v/v] TFA, 2% [v/v] ACN for analysis.

## Phosphopeptides enrichment for mass spectrometry

Twenty-four hours transfected HeLa cells or 24 h tetracycline (1 µg/mL) induced Flp-In T-REx 293 stable cells lines were harvested. Whole cell pellets or mitochondria pellets were solubilized, alkylated, and reduced at 2 µg/µL in SDC buffer (4% [w/v] SDC, 100 mM Tris pH 8.1, 40 mM CAA, 10 mM TCEP). Samples were boiled at 99 °C (5 min, 1,500 RPM), followed by 15 min sonication. Centrifugation at 21,000g (5 min, 4 °C) clarified the samples and the resultant supernatant was precipitated in final 65% [v/v] acetone, 20% [v/v] methanol at −20 °C overnight. The precipitates were pelleted with 16,100g (10 min, 4 °C) and washed once with 100% [v/v] acetone. The pellets were then solubilized at 5 µg/µL in 6

M guanidine HCl (Sigma-Aldrich), 50 mM Tris, 5 mM TCEP with sonication (20 min, RT). The samples were diluted to 1 M guanidine HCl with 50 mM Tris (pH 8) and all replicates are normalized to the same concentration and volume before digested with trypsin (protein:trypsin = 50:1) overnight at 37 °C.

SPE cartridges (Oasis # WAT094226) were washed once with 80% [v/v] ACN, 0.1% [v/v] TFA, followed by two washes with 0.1% [v/v] TFA. Trypsinized samples clarified with 21,000*g* (10 min, RT) were acidified to 1% [v/v] formic acid before loaded onto the pre-equilibrated cartridges. The cartridges were then washed two times with 0.1% [v/v] TFA before eluted with 80% [v/v] ACN, 0.1% [v/v] TFA. Twenty micrograms equivalent of each eluate were dried with CentriVap Benchtop Vacuum Concentrator (Labconco) and reconstituted in 0.1% [v/v] TFA, 2% [v/v] ACN for proteomics analysis. The rest of the eluates were freeze-dried with Christ Alpha 3–4 LSCbasic prior to phosphopeptides enrichment.

Titanium dioxide beads (TiO$_2$; GL Sciences #5210-21315) were washed three times with washing buffer (50% [v/v] ACN, 5% [v/v] TFA) at 1,000 rpm (10 min, RT), followed by two washes (1,000 rpm, 5 min, RT) with loading buffer (2 M lactic acid, 5% [v/v] TFA, 50% [v/v] ACN). TiO$_2$ beads were then equilibrated with loading buffer for at least 10 min (1,000 rpm, RT). Freeze-dried peptides were reconstituted at 1 µg/µL with loading buffer (15 min sonication, RT) and clarified with 21,000*g* (5 min, RT). Pre-equilibrated beads were added to the samples with TiO$_2$:peptides ratio of 6:1 for a 1 h incubation (1,000 rpm, RT). The suspensions were passed through stage-tips made in-house with 3 M Empore C8 extraction disks with centrifugation at 1,000 *g*. The flowthroughs were loaded back to the stage-tips once more to maximize binding of phosphopeptides binding to TiO$_2$. The TiO$_2$ columns were washed twice with loading buffer, followed by three washes with washing buffer, before three elutions with elution buffer (1% [v/v] ammonium hydroxide) and a final elution with 30% [v/v] ACN. The eluates were acidified to 10% [v/v] formic acid before freeze-dried and reconstituted in 2% [v/v] ACN, 0.05% [v/v] TFA for phosphopeptides analysis.

## LC MS/MS

LC MS/MS analysis of were performed either on an Orbitrap Eclipse, Orbitrap Ascend or Exploris 480 Mass Spectrometer (ThermoFisher Scientific). The LC systems were equipped with an Acclaim Pepmap nano-trap column (Dinoex-C18, 100 Å, 75 µm × 2 cm) and an Acclaim Pepmap RSLC analytical column (Dinoex-C18, 100 Å, 75 µm × 50 cm). The tryptic peptides were injected into the enrichment column at an isocratic flow of 5 µL/min of 2% [v/v] CH3CN containing 0.05% [v/v] TFA for 6 min, applied before the enrichment column was switched in-line with the analytical column. The eluents were 0.1% [v/v] FA (solvent A) in water and 100% [v/v] CH3CN in 0.1% [v/v] FA (solvent B), both supplemented with 5% DMSO. The gradient was at 300 nL/min from (i) 0–6 min, 3% B; (ii) 6–7 min, 3%–4% B; (iii) 7–82 min, 4%–25% B; (iv) 82–86 min, 25%–40% B; (v) 86–87 min, 40%–80% B; (vi) 87–90 min, 80%–3% B; (vii) 90–90.1 min, 80%–3% B and equilibrated at 3% B for 10 min before injecting the next sample.

For DDA experiments on an Orbitrap Eclipse, the mass spectrometer was operated in the data-dependent acquisition mode, whereby full MS1 spectra were acquired in a positive mode at 60,000 resolution. The 'top speed' acquisition mode with 3 s cycle time on the most intense precursor ion was used, whereby ions with charge states of 2–7 were selected. AGC target was set to standard with auto maximum injection mode. The 'top speed' acquisition mode with 3 s cycle time on the most intense precursor ion was used, whereby ions with charge states of 2–7 were selected. MS/MS analyses were performed by 1.6 *m/z* isolation with the quadrupole, fragmented by HCD with collision energy of 30%. MS2 resolution was at 15,000. Dynamic exclusion was activated for 30 s. AGC target was set to standard with auto maximum injection mode. Dynamic exclusion was activated for 30 s.

For DDA experiments on an Orbitrap Exploris 480, the mass spectrometer was operated in the data-dependent acquisition mode, whereby full MS1 spectra were acquired in a positive mode at 60,000 resolution. The 'top speed' acquisition mode with 3 s cycle time on the most intense precursor ion was used, whereby ions with charge states of 2–7 were selected. AGC target was set to standard with auto maximum injection mode. MS/MS analyses were performed by 1.6 *m/z* isolation with the quadrupole, fragmented by HCD with collision energy of 30%. MS2 resolution was at 15,000 Dynamic

exclusion was activated for 30 s. AGC target was set to standard with auto maximum injection mode. Dynamic exclusion was activated for 30 s.

For DIA experiments on an Orbitrap Ascend, full MS resolutions were set to 120,000 at $m/z$ 200 and scanning from 350 to 1400 $m/z$ in the profile mode. Full MS AGC target was 250% with an IT of 50 ms. AGC target value for fragment spectra was set at 2000%. Fifty windows of 13.7 Da were used with an overlap of 1 Da. Resolution was set to 30,000 and maximum IT to 55 ms. Normalized collision energy was set at 30%. All data were acquired in centroid mode using positive polarity.

## LC MS/MS data analysis

Raw mass spectra from co-immunoprecipitation were searched against a concatenated Uniprot human proteome database (canonical and reviewed, downloaded March 2021) and Uniprot fasta sequence of Lpg1625 (downloaded August 2023) using MaxQuant (version 2.1.4.0 and 2.5.2.0) [96]. Default label-free quantification (LFQ) parameters and match between runs were applied for the searches. The output "ProteinGroups.txt" was processed statistically with Perseus (version 1.6.10.43 and 2.0.5.0). Proteins that were "only identified by site", "reverse", and "contaminant" were removed from the dataset. The LFQ intensities were $Log_2$-transformed and grouped according to the experimental groups ($n = 3$). The data was filtered for ≥2 valid values in the experimental group only (e.g., FLAGLpg1625), and missing values in the control group (e.g., WT) were imputed from normal distribution (width = 0.3, and down shift = 1.8). Unpaired $t$ tests with FDR 0.05 were performed and data was displayed as volcano plots. Mitochondrial proteins and/or ER proteins were annotated with MitoCarta3.0 database (downloaded October 2023) [55] and Human Protein Atlas (proteins with main subcellular localization to ER, downloaded July 2024) [97].

For proteome and phosphoproteome analyses, their raw mass spectra were searched by the directDIA algorithm on Spectronaut 19 (version 19.0.240604) and against the Uniprot human proteome database (canonical and reviewed, downloaded March 2021). Default BGS Factory Settings and BGS Phospho PTM Workflow were used for directDIA searches of global proteome and phosphoproteome respectively, except that "Precursor PEP Cutoff", "Protein Qvalue Cutoff (Run)", and "Protein PEP Cutoff" were set to 0.01. The exported reports from Spectronaut were processed on Perseus (version 1.6.10.43 and 2.0.5.0) similar as above. PG.Quantity and PTM.Quantity from the proteomics and phosphoproteomics datasets respectively were $Log_2$-transformed and grouped according to the experimental groups ($n = 3$ or 4). Proteomics data were filtered for ≥ $n − 1$ valid values in each group before unpaired two-sample $t$ test with FDR 0.05 were performed and data presented as a scatterplot. Phosphoproteomics data was filtered to include only phosphopeptides with serine/threonine modified, and for ≥ $n − 1$ valid values in at least one group. Missing values in each group were imputed from normal distribution (width = 0.3, and down shift = 1.8) before unpaired two-sample $t$ test with FDR 0.05 were performed and data presented as a scatterplot.

## pLogo sequence motif analysis

"PTM.FlankingRegion" derived from the Spectronaut searches for phosphopeptides of interest was used as the input for the "foreground" (fg) and human protein was selected as the "background" (bg) for the analyses on https://plogo.uconn.edu/ (accessed October 2024).

## Supporting information

**S1 Fig. Lpg1625 is not required for intracellular replication of *Legionella pneumophila*.** Differentiated THP-1 cells were infected with *L. pneumophila* JR32 (wild-type), Δ*lpg1625*, Δ*lpg1625* complemented with FLAGLpg1625 or FLAGRVxF-dead (FLAGLpg1625K17A/V19A, mutant that loses interaction with PP1) with multiplicity of infection (MOI) of 5 ($n = 5$ independent experiments). The mean ± SEM for fold change of the colony-forming units (CFU/mL) of the different *L. pneumophila*

strains from THP-1 cells infected over three days against 3 hpi were plotted. Two-way ANOVA was used for statistical analysis. Corresponding raw data are provided in S2 Table.
(TIFF)

**S2 Fig. Localization specificity of <sup>FLAG</sup>Lpg1625 and charge mutants. (A)** HeLa cells transiently transfected with <sup>FLAG</sup>Lpg1625 were stained with 100 nM MitoTracker Deep Red FM (mitochondria; MTDR) and immunostained with antibodies against FLAG (green) and different organellar markers (magenta): NDUFAF2 (mitochondria), PDI (ER), PEX14 (peroxisomes), and GM130 (Golgi). Nuclei were stained with Hoechst 33258 (blue). Scale bar represents 10 μm. **(B)** The colocalization between <sup>FLAG</sup>Lpg1625 and different organellar markers was quantified using Pearson correlation coefficient ($r$) ($n = 3$ independent experiments, 5 cells each). Data represent mean ± SEM of the three experiments, with representative images presented in S2A Fig. Ordinary one-way ANOVA was performed on the means, comparing them to mitochondrial colocalization. Colocalization of <sup>FLAG</sup>Lpg1625 with all organellar markers was significantly lower than that with mitochondria, with a $p$-value < 0.0001. Corresponding raw data are available in S1 Data. **(C)** Colocalization between PDI and MTDR/<sup>FLAG</sup>Lpg1625 was quantified using Pearson correlation coefficient ($r$) ($n = 3$ independent experiments, 5 cells each). Data represents mean ± SEM of the three experiments, with representative images presented in S2A Fig. Unpaired $t$ test was performed on the means, showing no significant difference. Corresponding raw data are available in S1 Data. **(D)** The colocalization between PEX14 and MTDR/<sup>FLAG</sup>Lpg1625 was quantified using Pearson correlation coefficient ($r$) ($n = 3$ independent experiments, 5 cells each). Data represents mean ± SEM of the three experiments, with representative images presented in S2A Fig. Unpaired $t$ test was performed on the means, showing no significant difference. Corresponding raw data are available in S1 Data. **(E)** HeLa cells transiently transfected with <sup>FLAG</sup>Lpg1625<sup>K128A</sup> and <sup>FLAG</sup>Lpg1625<sup>AAA</sup> (K97A/R100A/K128A) were stained with 100 nM MitoTracker Deep Red FM (mitochondria; MTDR) and immunostained with antibodies against FLAG (green) and PDI (ER; magenta). Nuclei were stained with Hoechst 33258 (blue). Scale bar represents 10 μm. **(F)** Carbonate extraction of mitochondria isolated from HeLa cells transiently transfected with <sup>FLAG</sup>Lpg1625<sup>85-130</sup>. Corresponding raw images are available in S1 Raw Images.
(TIFF)

**S3 Fig. Validation of Flp-In T-REx 293 stable cell lines of <sup>FLAG</sup>Lpg1625 and <sup>FLAG</sup>RVxF<sub>dead</sub>. (A)** Titration of mitochondria isolated from tetracycline-induced Flp-In T-REx 293 stable cells expressing <sup>FLAG</sup>Lpg1625 and from HeLa cells transiently transfected with <sup>FLAG</sup>Lpg1625. Coomassie brilliant blue staining was used as a loading control. Corresponding raw images are available in S1 Raw Images, **(B)** Mitochondria sub-fractionation with mitochondria isolated from Flp-In T-REx 293 stable cell lines of <sup>FLAG</sup>Lpg1625. Isolated mitochondria were either left intact (lane 1 and 2), subjected to hypotonic swelling (lane 3 and 4), or solubilized with 0.5% Triton X-100 (lane 5 and 6). These samples were either left untreated or treated with 50 μg/mL Proteinase K (PK) and analyzed with SDS-PAGE and immunoblotting with the indicated antibodies. Corresponding raw images are available in S1 Raw Images. **(C)** Volcano plot showing proteins enriched in FLAG co-immunoprecipitation from mitochondria isolated from Flp-In T-REx 293 stable cell lines expressing <sup>FLAG</sup>Lpg1625 compared to wild-type Flp-In T-REx 293. The $\text{Log}_2$ fold change of mean LFQ intensity is plotted against $-\text{Log}_{10}$ $p$-value ($n = 3$ technical replicates). The curve indicates significantly enriched proteins (FDR = 0.05, s0 = 0.2). Mitochondrial outer membrane proteins are labeled with the annotation from MitoCarta3.0. Corresponding data are available in S4 Table. **(D)** Mitochondria sub-fractionation on mitochondria isolated from Flp-In T-REx 293 stable cell line of <sup>FLAG</sup>RVxF<sub>dead</sub> was performed as described in S3B Fig. Corresponding raw images are available in S1 Raw Images. **(E)** Volcano plot showing proteins enriched in FLAG co-immunoprecipitation samples of mitochondria isolated from Flp-In T-REx 293 stable cell lines of <sup>FLAG</sup>RVxF<sub>dead</sub> compared to wild-type Flp-In T-REx 293. The $\text{Log}_2$ fold change of mean LFQ intensity is plotted against $-\text{Log}_{10}$ $p$-value ($n = 3$ technical replicates). The curve indicates significantly enriched proteins (FDR = 0.05, s0 = 0.3). Mitochondrial outer membrane proteins are labeled with the annotation from MitoCarta3.0. Corresponding data are available in S4 Table.
(TIFF)

**S4 Fig. Substrates dephosphorylated in the presence of** $^{FLAG}Lp$**PIP or** $^{FLAG}Lp$**PIP-CYB5A**$^{TA}$**. (A)** Colocalization between $^{FLAG}Lp$PIP/$^{FLAG}Lp$PIP$^{K128A}$/$^{FLAG}Lp$PIP$^{AAA}$/$^{FLAG}Lp$PIP-CYB5A$^{TA}$ and mitochondria/ER was quantified using Pearson correlation coefficient (*r*) (*n* = 3 independent experiments, 5 cells each). Data represent mean ± SEM of the three experiments, with representative images presented in Figs 5A and S2E. Ordinary two-way ANOVA was performed on the means, comparing them to $^{FLAG}Lp$PIP within each group (MTDR or PDI). Colocalization of $^{FLAG}Lp$PIP with mitochondria (MTDR) was significantly higher than all the other variants, while colocalization of $^{FLAG}Lp$PIP with ER (PDI) was significantly lower than all the other variants. Corresponding raw data are available in S1 Data. **(B)** Representative images for Fig 5B. Untransfected HeLa cells and cells transiently transfected with $^{FLAG}Lp$PIP-CYB5A$^{TA}$ were immunostained with antibodies against FLAG (green)/PDI (ER; green) and PPP1CB (magenta). Nuclei were stained with Hoechst 33258 (blue). Scale bar represents 10 µm. **(C)** All proteins with phosphopeptides significantly reduced in abundance in HeLa cells transfected with $^{FLAG}Lp$PIP compared to empty vector pCDNA5 (Fig 5E) were used for gene ontology (GO) cellular compartment enrichment analysis with ShinyGO 0.80 [57]. The top 10 GO biological process terms are shown. Corresponding data are available in S2 Data. **(D)** Proteins with phosphopeptides significantly reduced in abundance in HeLa cells transfected with $^{FLAG}Lp$PIP-CYB5A$^{TA}$ compared to empty vector pCDNA5 (Fig 5F) were used for gene ontology (GO) cellular compartment enrichment analysis with ShinyGO 0.80 [57]. The top 10 GO biological process terms were shown. Corresponding data are available in S2 Data. **(E)** pLogo [56] sequence motif visualization of seven residues flanking 115 pSer/pThr sites found dephosphorylated in the presence of $^{FLAG}Lp$PIP, as shown in Fig 5E. Arginine (R) (log-odds = 7.182, *p*-value = 1.843 e−5) and proline (P) (log-odds = 10.779, *p*-value = 4.653 e−9) at position −3 and +1 relative to pSer/pThr respectively were found to be significantly overrepresented. **(F)** pLogo [56] sequence motif visualization of seven residues flanking 351 pSer/pThr sites found dephosphorylated in the presence of $^{FLAG}Lp$PIP-CYB5A$^{TA}$ as shown in Fig 5F. Arginine (R) (log-odds = 11.020, *p*-value = 2.674 e−9) and proline (P) (log-odds = 33.621, *p*-value = 6.700 e−32) at position −3 and +1 relative to pSer/pThr respectively were found to be significantly overrepresented. **(G)** Proteins with phosphopeptides significantly increased in abundance in HeLa cells transfected with $^{FLAG}Lp$PIP compared to empty vector pCDNA5 (Fig 5E) were used for gene ontology (GO) cellular compartment enrichment analysis with ShinyGO 0.80 [57]. The top 10 GO biological process terms are shown. Corresponding data are available in S2 Data. **(H)** Proteins with phosphopeptides significantly increased in abundance in HeLa cells transfected with $^{FLAG}Lp$PIP-CYB5A$^{TA}$ compared to empty vector pCDNA5 (Fig 5F) were used for gene ontology (GO) cellular compartment enrichment analysis with ShinyGO 0.80 [57]. The top 10 GO biological process terms were shown. Corresponding data are available in S2 Data.
(TIFF)

**S1 Table. Bioinformatic screen for TA proteins from *Legionella pneumophila* T4SS effectors.**
(XLSX)

**S2 Table. Raw data for the quantification of fold change in colony forming units (CFU/mL) of WT, Δ*lpg1625*, Δ*lpg1625*/**$^{FLAG}$**Lpg1625, and Δ*lpg1625*/**$^{FLAG}$**RVxF**$_{dead}$ ***Legionella pneumophila* strain JR32 from THP-1 cells infected with a MOI of 5 over 3 days (see S1 Fig).**
(XLSX)

**S3 Table. Co-immunoprecipitation datasets from HeLa cells transfected with** $^{FLAG}$**Lpg1625 and** $^{FLAG}$**RVxF**$_{dead}$ **(see Fig 2A and 2D).**
(XLSX)

**S4 Table. Co-immunoprecipitation datasets from isolated mitochondria of tetracycline-induced Flp-In T-REx 293 stable cells expressing** $^{FLAG}Lp$**PIP and** $^{FLAG}$**RVxF**$_{dead}$ **(see S3C and S3E Fig).**
(XLSX)

**S5 Table. Mitochondrial proteome and phosphoproteome datasets from wild-type, FLAGLpPIP and FLAGRVxF$_{dead}$ Flp-In T-REx 293 stable cell lines (see Fig 3B–3D).**
(XLSX)

**S6 Table. Co-immunoprecipitation datasets from HeLa cells transfected with FLAGLpPIP or FLAGLpPIP-CYB5A$^{TA}$ (see Fig 5C and 5D).**
(XLSX)

**S7 Table. Whole cell global proteome and phosphoproteome datasets from HeLa cells transfected with FLAGLpPIP, FLAGLpPIP-CYB5A$^{TA}$, or empty vector (pCDNA5) (see Fig 5E and 5F).**
(XLSX)

**S1 Text. Experimentally validated or putative *Legionella pneumophila* T4SS effectors.**
(DOCX)

**S2 Text. *Legionella pneumophila* T4SS effectors containing an RVxF motif.**
(DOCX)

**S1 Data. Raw data for quantitative analyses derived from microscopy images.**
(XLSX)

**S2 Data. GO enrichment analysis of proteins with significantly upregulated or downregulated phosphopeptides, performed using ShinyGO 0.80.**
(XLSX)

**S3 Data. Raw data for OCR and ECAR (see Fig 4A and 4B).**
(XLSX)

**S4 Data. Densitometric quantification of [$^{35}$S]-Tim23 and [$^{35}$S]-GC1 in vitro import (see Fig 4H and 4I).**
(XLSX)

**S1 Raw Images. Raw images for western blots and autoradiographs.**
(PDF)

## Acknowledgments

We thank Prof Roger Daly for insightful discussions on PP1 activity, and Prof Michael Lazarou for input on the mt-Keima assay. We thank the Biological Optical Microscopy Platform and the Mass Spectrometry and Proteomics Facility at the Bio21 institute, as well as the Melbourne Cytometry Platform at the Peter Doherty Institute, for access to instrumentation, training, and technical support.

## Author contributions

**Conceptualization:** Hayley J. Newton, Diana Stojanovski.

**Data curation:** Kai-Qi Yek, Ching-Seng Ang, Catherine S. Palmer, Ann E. Frazier.

**Formal analysis:** Kai-Qi Yek, Evie R. Hodgson, Ching-Seng Ang, Catherine S. Palmer, Ann E. Frazier.

**Funding acquisition:** Hayley J. Newton, Diana Stojanovski.

**Investigation:** Kai-Qi Yek, Evie R. Hodgson, Ching-Seng Ang, Diana Stojanovski.

**Methodology:** Kai-Qi Yek, Evie R. Hodgson, Ching-Seng Ang.

**Project administration:** Hayley J. Newton, Diana Stojanovski.

**Resources:** Diana Stojanovski.

**Supervision:** Hayley J. Newton, Diana Stojanovski.

**Writing – original draft:** Kai-Qi Yek, Hayley J. Newton, Diana Stojanovski.

**Writing – review & editing:** Kai-Qi Yek, Evie R. Hodgson, Ching-Seng Ang, Catherine S. Palmer, Ann E. Frazier, Hayley J. Newton, Diana Stojanovski.

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
