## [Editor Report · Decision Letter 0]

Dear Diana,

Thank you for submitting your manuscript entitled "Protein phosphatase 1 (PP1) drives dephosphorylation of mitochondrial outer membrane proteins via the Legionella effector protein, LpPIP" for consideration as a Research Article by PLOS Biology.

Your manuscript has now been evaluated by the PLOS Biology editorial staff as well as by an academic editor with relevant expertise and I am writing to let you know that we would like to send your submission out for external peer review.

Once your full submission is complete, your paper will undergo a series of checks in preparation for peer review. After your manuscript has passed the checks it will be sent out for review. To provide the metadata for your submission, please Login to Editorial Manager (https://www.editorialmanager.com/pbiology) within two working days, i.e. by Nov 14 2024 11:59PM.

Kind regards,

Ines

--

Ines Alvarez-Garcia, PhD

Senior Editor

PLOS Biology

---

## [Decision Letter · Decision Letter 1]

Dear Dr Stojanovski,

Thank you for your patience while your manuscript entitled "Protein phosphatase 1 (PP1) drives dephosphorylation of mitochondrial outer membrane proteins via the Legionella effector protein, Lp PIP" was peer-reviewed at PLOS Biology and please accept my apologies for the delay in sending you our decision. The manuscript has been evaluated by the PLOS Biology editors, an Academic Editor with relevant expertise, and by three independent reviewers.

The reviews are attached below. As you will see, the reviewers find the conclusions interesting and novel, but they also raise several points that would need to be addressed before we can consider the manuscript for publication. Reviewer 1 thinks it’s unclear what is the downstream consequence of the dephosphorylation on mitochondrial biology and cell survival, and that it would help to show that not all organelles show a similar overlap with Lpg1625-FLAG staining as a control. This reviewer also asks for several clarifications and a discussion of the limitations of the study. Reviewer 2 mentions that you could also test whether Lpg1625 also targets peroxisomes and perform a co-staining with an ER marker to confirm that the Lpg625 charge variants are targeted to the ER. In addition, this reviewer suggests checking if the C-tail is sufficient for mitochondrial targeting and membrane integration by subcellular fractionation. Finally, Reviewer 3 makes several suggestions for improvement, including checking if there are quantifiable differences in morphology between wild type, LpPIP and RVxF dead cells.

In light of the reviews, we would like to invite you to revise the work to thoroughly address the reviewers' reports. Given the extent of revision needed, we cannot make a decision about publication until we have seen the revised manuscript and your response to the reviewers' comments. Your revised manuscript is likely to be sent for further evaluation by all or a subset of the reviewers.

**IMPORTANT - SUBMITTING YOUR REVISION**

3. Resubmission Checklist

a) *PLOS Data Policy*

b) *Published Peer Review*

Sincerely,

Ines

--

Ines Alvarez-Garcia, PhD

Senior Editor

PLOS Biology

Reviewers' comments

Rev. 1:

Summary:

The authors identify Lpg1625 as a mitochondrial-targeted effector of Legionella pneumophilia which interacts with the protein phosphatase 1 (PP1). They offer evidence that Lpg1625 interacts with PP1 via a specific RVxF motif, recruiting the phosphatase to mitochondria where it will subsequently dephosphorylate mitochondrial proteins. The proteins targeted by the recruited PP1 are primarily involved in mitochondrial dynamics, potentially linking it to the effects that Legionella pneumophilia infection has on the host. Mistargeting of the protein to the ER changes its substrate specificity to ER proteins, indicating that it is primarily the localization of the phosphatase that elicits its specific effect on mitochondrial proteins.

This paper presents well conducted experiments that support the proposed model and showcase an interesting mechanism for the organellar recruitment of PP1 that is exploited by Legionella. However, it remains unclear what the benefit of the recruitment for the bacterium is. Furthermore, some smaller inconsistencies need to be remedied (see below). I therefore suggest to accept this paper after some revisions.

Major concerns:

- While this paper nicely showcases the effect of Lpg1625 on the localization of PP1 and the subsequent dephosphorylation of mitochondrial proteins, it is unclear what the downstream consequence of this dephosphorylation is on mitochondrial biology and cellular survival. The authors discuss PP1 action on Bcl-2 and the role of FUNDC1 in cell death, but does the expression of Lpg1625 alter cell survival? Or mitochondrial parameters such as respiration or morphology?

- While the evidence presented supports a mitochondrial localization, it would be good to show that not all organelles show a similar overlap with Lpg1625-FLAG staining. This should also be quantified.

- The authors argue that, when recruited to mitochondria, PP1 is active and dephosphorylates specific targets. Does this reduce the PP1 activity at other subcellular sites? What fraction of PP1 is recruited to mitochondria?

- The authors claim that binding to Lpg1625 does not prevent PP1 activity because it occurs outside of the catalytic site. Can they rule out allosteric inhibition?

- For Fig. 1E, the authors argue that localization of the K128A and K97A/R100A/K128A mutants "showed localization reminiscent of the ER" yet they did not co-transfect any ER marker which would prove this association; could this be added?

- The discussion section would benefit from the authors also discussing the limitations of their experiments / the model that they are suggesting.

Minor concerns / comments:

- Labelling of the y-axis in Fig S1 does not completely match the figure legend.

- In Fig S1, the authors make use of a FLAGRVxFdead mutant yet creation of this mutant is only explained for Fig. 2C. I would have preferred to see this explanation earlier in the text.

- Following Fig 2, the authors switch from HeLa to HEK293 cells. Could they please explain the reasoning behind this?

- For Fig. 3D, the authors list n = 4 and n = 3. Is this the number of cells or experimental repeats?

- The authors argue that Fig S2A confirms Fig 1D regarding the localization of Lpg1625 on the outer mitochondrial membrane. In Fig S2A however there is an additional faint band in the upper panel under treatment with PK. This band does not show up in Fig 1D. Do the authors know what is labelled here (e.g. is it a known unspecific band or something different)?

- The authors switch between American and British English (sometimes even within the same sentence)

- The authors are inconsistent in their usage of chemical units: ml vs mL

- The authors are inconsistent regarding their use of tenses: e.g. p. 17, line 422-424: "primary antibodies were" vs "secondary antibodies used are"

Rev. 2:

Many bacterial pathogens use dedicated translocation systems to deliver effector proteins to their hosts. Legionella pneumophila has the largest known bacterial arsenal of effector proteins (>300). In the present contribution the authors identified one of these effector (Lpg1625) proteins to be a tail-anchored protein embedded in the outer mitochondrial membrane. They further showed that Lpg1625 interacts with human protein phosphatase (PP1), a highly conserved serine/threonine phosphatase and therefore they renamed the protein to LpPIP. The LpPIP-mediated recruitment of PP1 to the surface of mitochondria affect the phosphorylation pattern of proteins residing in the mitochondrial outer membrane.

Major points

1. Several mitochondrial TA proteins (like MFF and FIS1) are dually localized to peroxisomes. It will be interesting to test whether Lpg1625 targets also peroxisomes. For example, can it be that the small puncta structures in Fig. 1C, which are not co-stained with the mitochondrial marker, represent peroxisomal structures?

2. Fig. 1E: (i) Although possible, the evidence that the charge variants of Lpg1625 are targeted to the ER is not convincing. A co-staining with an ER marker is required. (ii) To support the proposal that the C-tail of Lpg1625 is sufficient for mitochondrial targeting and membrane integration, it will be helpful to show by biochemical means (subcellular fractionation) that this construct is enriched in mitochondrial fraction.

3. Fig. 2E: To support the assumption that PPP1CB (or other subunits of PP1) associates with mitochondria upon expression of LpPIP, it will be good to show by western blotting that PP1 is found on isolated mitochondria under these conditions.

4. The authors write in lines 221-2 that "This suggests that LpPIP-PP1 holoenzyme could be modulating mitochondria dynamics." To place their finding in a physiological context, the authors should compare mitochondrial morphology in cells harboring LpPIP to cells expressing its FLAG-RVxFdead variant.

Minor points:

A. Among the four TA proteins, was there a special reason for selecting Lpg1625 as the first candidate for further functional analysis?

B. Line 139: As far as I recall, there is no evidence for the involvement of Tom40 in the biogenesis of mitochondrial TA proteins. Thus, the authors might consider rephrasing their text or cite an appropriate reference. Along the same line, if TOM40 is involved in the biogenesis of Lpg1625, it should also interact with the FLAG-RVxFdead variant (Fig. 2D)

C. Fig. S2A and S2C: The lanes representing the samples with PK contain a band which migrate slightly slower than FLAG-Lpg1625. The authors should explain this band.

D. Line 164: Is Lys16 part of the RVxF motif? The authors should explain this point.

E. Lines 245-7: The effect on phosphorylation of nuclear proteins was observed upon overexpression of LpPIP. It will be good if the authors will test this issue also upon lower (and maybe more physiological) expression levels of LpPIP.

F. What is the difference between Fig. 3E and the panel in Fig. S3E. Please explain why they are not identical.

Rev. 3:

The mitochondrion is a key organelle targeted by pathogens that aim to hijack cellular processes. Legionella pneumophila translocates ~300 effector proteins into host cells using a type IVB secretion system to achieve this. In this manuscript, Yek et. al. sought to identify Legionella effector proteins that modify mitochondrial behavior through localizing to the outer mitochondrial membrane (OMM) using a tail anchor (TA), a common feature of some OMM proteins. Using a bioinformatic approach, 4 candidate effector TA domain-containing proteins were confirmed to localize to mitochondria in mammalian cells. One of these effectors, Lpg1625, or LpPIP, was selected for further molecular characterization. The authors then used biochemical and immunofluorescent approaches to confirm that localization of LpPIP to the OMM is dependent on its transmembrane domain and positively charged C-tail. The authors next sought to identify a molecular function for the effector protein LpPIP through identifying interacting partners using an affinity enrichment mass spectrometry approach in transient overexpression and stable doxycycline inducible cell lines and discovered the Ser/Thr phosphatase PP1 as a strong interactor. Using AE-MS and immunofluorescence, they then confirmed PP1 interaction with and recruitment to mitochondria by LpPIP was dependent on an RVxF motif in its N-terminus, a motif commonly found in PP1 interacting proteins.

As PP1 is a ubiquitously expressed phosphatase, the authors chose to examine the phosphoproteomic profile of HEK293 cells overexpressing WT or RVxF dead version of LpPIP and identified a number of mitochondrial phosphoproteins whose abundances decreased in WT cells only, indicating that PP1 is likely mediating dephosphorylation of OMM proteins in a manner dependent on its recruitment by LpPIP. Subsequent bioinformatic analyses suggest that there is an overrepresentation of phosphopeptides involved in mitochondrial dynamics which contain an RXXpS motif, suggesting that LpPIP mediates the recruitment of PP1 to the OMM and enables the dephosphorylation of proteins involved in mitochondrial fission and fusion. The authors confirmed that the TA signature of LpPIP is important for directing PP1 to the OMM, and is not involved in mediating PP1 substrate specificity, through demonstrating that ER-localized LpPIP (LpPIP-CYB5A) promotes PP1 localization to the ER, and dephosphorylation of ER-specific proteins only.

Overall, I believe the experiments are designed and executed well and provide interesting insights into the modulation of mitochondrial phosphorylation by a pathogen effector protein. The paper is well written, and the conclusions drawn interpret the data appropriately. Below I have provided some comments that could strengthen the manuscript.

* Mitochondrial morphology of transiently expressed effector proteins Lpg1803 and Lpg2344 do not appear to overlap exclusively with NDUFAF2, suggesting either non-mitochondrial localization, or disruption to mitochondrial integrity. Expression of Lpg1625 (LpPIP), Lpg1803, and Lpg2344 appears to impact overall mitochondrial morphology. Given the identification of mitochondrial dynamics as an enriched GO term for phosphoproteins regulated in response to LpPIP expression, the authors could examine if there are quantifiable differences in morphology between wild type, LpPIP and RVxF dead cells. Additionally, higher magnification images should be shown to support the conclusions drawn in the manuscript.

* The authors should provide representative images that were used in generating the results shown in Fig. 4B.

* The changes in mitochondrial protein phosphorylation suggest altered functionality of, for example, protein import. Is there a measurable effect on this or any other process related to the phosphoproteins?

---

## [Decision Letter · Decision Letter 2]

Dear Dr Stojanovski,

Thank you for your patience while we considered your revised manuscript entitled "Protein phosphatase 1 (PP1) drives dephosphorylation of mitochondrial outer membrane proteins via the Legionella effector protein, LpPIP." for publication as a Research Article at PLOS Biology. This revised version of your manuscript has been evaluated by the PLOS Biology editors, the Academic Editor and the three original reviewers.

Based on the reviews, we are likely to accept this manuscript for publication, provided you satisfactorily address the remaining points raised by Reviewer 2. Please also make sure to address the data and other policy-related requests stated below my signature.

In addition, we would like you to consider a suggestion to improve the title:

"Legionella effector LpPIP recruits protein phosphatase 1 to the mitochondria to induce dephosphorylation of outer membrane proteins"

We expect to receive your revised manuscript within two weeks.

*Published Peer Review History*

*Press*

Sincerely,

Ines

--

Ines Alvarez-Garcia, PhD

Senior Editor

PLOS Biology

DATA POLICY:

I can see that you have some of the data on the tables, however we do require all individual quantitative observations that underlie the data summarized in the graphs shown in the figures and results of your paper be made available in one of the following forms:

Fig. 2A, D, E, F, G; Fig. 3B-D, F; Fig. 4A, B, C, E, G, H, I, J; Fig. 5A-F; Fig. S1; Fig. S2B-D; Fig. S3B, D and Fig. S4A, C, D, G, H

***In addition, please make the data you have deposited in the ProteomeXchange Consortium via the PRIDE (PXD057005) publicly available at this stage.

CODE POLICY

Reviewers' comments

Rev. 1:

The authors have answered all points raised sufficiently. I especially appreciate the detailed analysis of possible mitochondrial phenotypes, even though no direct effects were observed. The reasons for this are nicely explained in the discussion and will help the field to further evaluate the interplay between Legionella and its host.

Rev. 2:

The authors nicely addressed all my comments on the original submission.

There are only two minor points that the authors might consider before publication:

1. Fig. S2F: The PK treatment is not mentioned in the text and is also not convincing as the inner membrane protein TIM29 is also affected. I would suggest to remove this part from the figure.

2. Lines 154-156 and Fig. S3: the authors write: "To rule out potential artifacts, we generated a stable tetracycline-inducible Flp-InTMT-RExTM 293 cell line expressing FLAGLpg1625, allowing regulation of expression to low levels." However, they do not show that this cell line indeed expresses less FLAGLpg1625 as compared to the cells used in the previous assays (Figs. 1 and 2). A comparison by Western blot where both cell lines are analysed side-by-side will be helpful.

Rev. 3:

The authors have adequately addressed my critiques. I am happy to recommend publication.

---

## [Editor Report · Decision Letter 3]

Dear Dr Stojanovski,

Thank you for the submission of your revised Research Article entitled "Legionella effector LpPIP recruits protein phosphatase 1 to the mitochondria to induce dephosphorylation of outer membrane proteins." for publication in PLOS Biology. On behalf of my colleagues and the Academic Editor, Andre Schneider, I am delighted to let you know that we can in principle accept your manuscript for publication, provided you address any remaining formatting and reporting issues. These will be detailed in an email you should receive within 2-3 business days from our colleagues in the journal operations team; no action is required from you until then. Please note that we will not be able to formally accept your manuscript and schedule it for publication until you have completed any requested changes.

PRESS

Sincerely, 

Ines

--

Ines Alvarez-Garcia, PhD

Senior Editor

PLOS Biology
